# The sensitivity of atmospheric blocking to upstream latent heating - numerical experiments

Daniel Steinfeld[1], Maxi Boettcher[1], Richard Forbes[2], and Stephan Pfahl[3]

[1]Institute for Atmospheric and Climate Science, ETH Zürich, Switzerland
[2]European Centre for Medium-Range Weather Forecasts, Reading, United Kingdom
[3]Institute of Meteorology, Freie Universität Berlin, Germany

**Correspondence:** Daniel Steinfeld (daniel.steinfeld@alumni.ethz.ch)

**Abstract.** Recent studies have pointed to an important role of latent heating during cloud formation for the dynamics of anti-cyclonic circulation anomalies such as atmospheric blocking. However, the effect of latent heating on blocking formation and maintenance has not yet been fully elucidated. To explicitly study this cause-and-effect relationship, we perform sensitivity simulations of five selected blocking events with the IFS global weather prediction model in which we artificially eliminate la-
tent heating in clouds upstream of the blocking anticyclones. This elimination has substantial effects on the upper-tropospheric circulation in all case studies, but there is also significant case-to-case variability: some blocking systems do not develop at all without upstream latent heating, while for others the amplitude, size and lifetime of the blocking anticyclones are merely reduced. This strong influence of latent heating on the mid-latitude flow is due to the injection of air masses with low potential vorticity (PV) into the upper troposphere in strongly ascending "warm conveyor belt" airstreams, and the interaction of the
associated divergent outflow with the upper-level PV structure. The important influence of diabatic heating demonstrated with these experiments suggests that the accurate representation of moist processes in ascending airstreams in weather prediction and climate models is crucial for blocking dynamics.

**Keywords**: atmospheric blocking, atmospheric dynamics, jet stream, extratropical cyclone, mid-latitude weather, latent heating, diabatic processes, potential vorticity, numerical sensitivity simulation.

*Copyright statement.*

## 1   Introduction

The formation and maintenance of prolonged anticyclonic circulation anomalies, denoted as atmospheric blocking, represents an important and challenging aspect of mid-latitude weather variability. Atmospheric blocking leads to persistent changes in the large-scale circulation and blocks the westerly flow (Rex, 1950; Woollings et al., 2018), often causing anomalous, sometimes
extreme weather (Green, 1977) in a situation of increased forecast uncertainty in weather models (Pelly and Hoskins, 2003; Rodwell et al., 2013).

Despite its importance, there is currently no comprehensive theory of blocking (for a review see Tyrlis and Hoskins, 2008). Several dynamical processes have been identified to be conducive to blocking formation, such as planetary-scale wave dynamics (e.g., Charney and DeVore, 1979; Hoskins and Valdes, 1989; Petoukhov et al., 2013), forcing by transient eddies (e.g., Shutts, 1983; Luo et al., 2014) and Rossby wave breaking (Pelly and Hoskins, 2003; Altenhoff et al., 2008), with evidence that different processes can dominate in different blocking cases (Nakamura et al., 1997; Drouard and Woollings, 2018; Steinfeld and Pfahl, 2019). Atmospheric blocking occurs when an air mass with anomalously low potential vorticity (PV) is advected poleward, related to a meridionally amplified flow (Nakamura and Huang, 2018), setting up a large-scale negative (anticyclonic) PV anomaly in the upper troposphere at the level of the mid-latitude jet stream and a stable surface anticyclone underneath (Hoskins et al., 1985). Such large-scale advection of anticyclonic air masses into the blocking region occurs typically on the downstream side of developing baroclinic waves (e.g., Colucci, 1985; Mullen, 1987; Nakamura and Wallace, 1993; Yamazaki and Itoh, 2012), which is the synoptic mechanism behind the classical 'eddy-mean flow' view, i.e. the dynamical interaction between synoptic transient eddies and the large-scale blocked flow (Berggren et al., 1949; Green, 1977; Shutts, 1983; Hoskins et al., 1983).

While these studies focused on dry-adiabatic mechanisms and the isentropic advection of low PV air, recent case (Croci-Maspoli and Davies, 2009; Lenggenhager et al., 2019; Maddison et al., 2019) and climatological (Pfahl et al., 2015; Steinfeld and Pfahl, 2019) studies based on air parcel trajectory calculations demonstrated that moist-diabatic processes, and in particular latent heating (LH) during cloud formation in strongly ascending airstreams, play a significant role for the dynamics of blocking. The primary effect of latent heat release on blocking is the diabatic generation and amplification of upper-level negative PV anomalies (Pfahl et al., 2015). This amplification results from the injection of low PV into the upper troposphere in cross-isentropic ascending airstreams and the interaction of the diabatically enhanced divergent outflow with the upper-level PV structure at the tropopause (Steinfeld and Pfahl, 2019). For example, these diagnostic studies have shown that LH occurs predominantly in the warm conveyor belt (WCB; Wernli, 1997; Methven, 2015) of extratropical cyclones and is generally most important during blocking onset and in more intense and larger blocks. In addition, the repeated injection of diabatically heated low PV air during the blocking life cycle, associated with a series of transient cyclones approaching the block, can act to maintain blocks against dissipation. These findings complement the large body of previous work that found LH to be important for the development of mid-latitude weather systems, such as cyclones (Ahmadi-Givi et al., 2004; Binder et al., 2016), anticyclones (Quinting and Reeder, 2017; Zschenderlein et al., 2020), Rossby waves (Grams et al., 2011; Röthlisberger et al., 2018) and Rossby wave breaking (Zhang and Wang, 2018).

Nevertheless, as these previous studies have used diagnostic methods to determine statistical relationships between LH and blocking, this does not necessarily mean that LH has a strong causal impact on blocking. The effect of LH on blocking, and for Rossby wave dynamics at the tropopause in general, is still not completely understood. It is a challenge to quantify the impact of LH, mainly because LH is strongly coupled to the dry dynamics of baroclinic waves and the associated adiabatic advection of PV (e.g., Kuo et al., 1990; Teubler and Riemer, 2016). The question of whether LH critically modifies the development of blocking, that is otherwise mostly affected by dry dynamics, and the investigation of the corresponding cause-and-effect relationship is the focus of this study.

The main objective of this paper is to study the sensitivity of atmospheric blocking to changes in upstream LH in numerical model simulations. The effects of LH on the development (onset, maintenance and decay) of five different blocking events are studied in detailed model-based sensitivity experiments, in which cloud-related LH is altered in the storm track region upstream of the block, and compared to control simulations with unmodified upstream LH. In doing so, changes in the formation and maintenance of blocking in these simulations can be attributed to altered LH upstream.

The sensitivity experiments are presented as follows. Section 2 describes the methodology, while section 3 introduces one blocking event as an example with a synoptic overview. The results of the sensitivity experiments are presented in section 4 and our conclusions are summarized and discussed in section 5.

## 2 Methods

### 2.1 Model setup

This work is based on numerical simulations with ECMWF's global Integrated Forecast System (IFS) cycle 43R1, which was operational between November 2016 and July 2017. The model is run at a cubic spectral truncation of TCo319, which corresponds to roughly 32 km grid spacing, and with 91 vertical hybrid pressure–sigma levels with a vertical resolution in the upper troposphere of about 10–25 hPa (ECMWF, 2016a). ECMWF operational analysis fields are used for initial conditions. 3-hourly output fields, including physical temperature tendencies, are interpolated to a regular grid at 1° horizontal resolution.

In the IFS sub-grid scale processes are represented by various parametrization schemes (ECMWF, 2016b). The cloud and large-scale precipitation microphysics scheme, based on Tiedtke (1993), includes five prognostic variables (cloud fraction, cloud liquid water, cloud ice, rain and snow) with associated sources and sinks (Forbes et al., 2011; Ahlgrimm and Forbes, 2013; Forbes and Ahlgrimm, 2014). Convection is parametrized according to Tiedtke (1989) and Bechtold et al. (2008), with a modified CAPE closure (Bechtold et al., 2013).

### 2.2 Sensitivity experiments

The effect of cloud-diabatic heating on atmospheric blocking is investigated with sensitivity experiments by comparing the full-physics control simulation including LH (hereafter referred to as CNTRL) to the corresponding simulation without LH (NOLH). LH is artificially turned off by multiplying the instantaneous temperature tendencies due to parameterized cloud and convection processes with a factor $\alpha = 0.0$, but still allowing for moisture changes due to cloud and precipitation formation. Other non-conservative processes, such as radiative heating and turbulent mixing, which can also modify PV (Spreitzer et al., 2019; Attinger et al., 2019), are not altered.

In contrast to previous numerical sensitivity studies that investigated cyclone dynamics and modified LH everywhere in the model domain (e.g., Kuo et al., 1990; Stoelinga, 1996; Büeler and Pfahl, 2017), here LH is only modified in the region that is identified to be directly relevant for the blocking system, which is typically the WCB ascent region associated with upstream extratropical cyclones (Steinfeld and Pfahl, 2019). In doing so, we can attribute the changes in the structure of the blocking

in these simulations to the altered LH in the confined upstream region, while allowing for heating/cooling everywhere else in the global domain. To define the heating region objectively, location and time of strongest increase in potential tempera-
ture are determined along backward trajectories initiated in the upper-tropospheric blocking in the CNTRL simulation. Our experiment aims to suppress strongly ascending airstreams like WCBs that lead to a strong divergent outflow and the presence of diabatically modified PV at the tropopause. Heating along the WCB by cloud microphysical processes and convection is strongest in the lower and middle troposphere (Joos and Wernli, 2012; Oertel et al., 2019). In order to isolate the effect of this LH, a 3-dimensional box is placed over the main heating region, which is kept fixed during each NOLH simulation, and LH is only modified in this box. The box has a vertical extent between 900 - 500 hPa and a horizontal extent which is adjusted for each blocking case (see Table 1). It should be kept in mind that other microphysical processes, such as ice-phase microphysics close to the outflow level, can also contribute to the heating and PV modification along the WCB (Joos and Wernli, 2012). As these processes also occur above 500 hPa, our approach does not fully remove all cloud-related LH, and there is still moderate heating/cooling outside of the box. Near the edges of the box (in a zone of 5° horizontally and 50 hPa in the vertical), the temperature tendency multiplying factor alpha is interpolated linearly to obtain a smooth transition from $\alpha = 0.0$ to $1.0$.

The sensitivity experiments are performed for five selected case studies of blocking events (Table 1). The simulations are run for 10 days. We chose the initialization time for each case based on two requirements: (1) LH has to be removed early enough to ensure its contribution to the initial ridge amplification is minimal and (2) the CNTRL simulation needs to adequately simulate the development of the observed block, as verified visually against ECMWF analysis. For all cases, the simulations are initialized during the intensification phase of an upstream cyclone, which is typically between 2 - 3 days prior to blocking onset. The blocking decay is not always captured in the 10-day simulations, as many blocks persisted longer. Although a 10-day forecast simulation does not perfectly match observations/analysis (cf. supplemental Fig. S4), especially during blocking situations with increased forecast uncertainty (e.g., Tibaldi and Molteni, 1990; Pelly and Hoskins, 2003; Matsueda, 2009), such differences do not affect the conclusions obtained from the sensitivity experiments, since we compare simulations with LH (CNTRL) to simulations without LH (NOLH).

LH in extratropical cyclones is coupled to and interacts with other processes, and hence, its artificial removal can affect many aspects of the flow, such as the cyclone intensification and its baroclinic coupling to the upper-level trough (e.g., Hoskins et al., 1985; Ahmadi-Givi et al., 2004). The role of LH in explosively developing cyclones has been studied in great detail, and thus, we focus on the evolution and structure of upper-level blocking here. However, to better understand such non-linear interactions and their effect on the large-scale flow, we additionally conduct sensitivity experiments with reduced LH ($\alpha = 0.5$) and increased LH ($\alpha = 1.5$) for one specific blocking event.

## 2.3 Diagnostic methods

A combination of Eulerian and Lagrangian diagnostics is applied to study and quantify the processes involved in the development of blocking, and in particular the role of latent heat release in ascending airstreams. The term "upper-level" is used hereafter to describe the vertically averaged flow between 500 and 150 hPa.

### 2.3.1 Atmospheric blocking tracking

Following Schwierz et al. (2004a), blocking is identified and tracked as upper-level negative PV anomalies. The anomalies are calculated with respect to the calendar-month averages over the ERA-Interim reanalysis period 1979–2016 (Dee et al., 2011) and temporally smoothed with a 2-day running mean filter. Different thresholds for intensity, persistence and quasi-stationary have been tested in order to track and compare upper-level negative PV anomalies in both CNTRL and NOLH simulations. In all simulations, blocks are identified with a threshold of -1 pvu and a spatial overlap of 80 % between two consecutive time steps. No persistence criterion is applied. The reason for this is that the tracked negative PV anomalies in the NOLH simulations are weak (see below) and would not be classified as persistent blocks (see also Croci-Maspoli et al., 2007). Nevertheless, all blocking events investigated here also fulfill the stricter blocking criteria used, e.g., by Steinfeld and Pfahl (2019) in the CNTRL simulation. The advantage of the PV-anomaly-based index is that it objectively captures the core of the anomalous anticyclonic circulation and thus directly allows for an investigation of the origin and evolution of individual blocks and the associated air masses. A number of relevant blocking characteristics and their evolution are calculated during the blocking life cycle, such as location of the blocking center (center of mass), spatial extent, blocking intensity (area-averaged upper-level negative PV anomaly) and lifetime. The calculated quantities are area-weighted.

### 2.3.2 Effects of latent heating

To capture the full three-dimensional complexity of LH in ascending airstreams and to quantify its effect on blocking dynamics, a combined Eulerian and Lagrangian perspective is adapted. The effects of LH on the upper-tropospheric PV distribution are quantified as follows:

– **Backward trajectories**: To estimate the relative contributions of dry (quasi-adiabatic transport of mass) and moist (cross-isentropic transport of mass) processes to upper-level negative PV anomalies that characterize blocking, we compute kinematic 3-day backward air-parcel trajectories based on the three-dimensional wind using the Lagrangian Analysis tool LAGRANTO (Wernli and Davies, 1997; Sprenger and Wernli, 2015). The trajectories are started from an equidistant grid ($\Delta x = 100$ km horizontally and $\Delta p = 50$ hPa vertically between 500 and 150 hPa) in the blocking region every three hours, with the additional criterion that PV must be smaller than 1 pvu to exclude points located in the stratosphere. Since both PV and potential temperature $\theta$ are conserved for adiabatic and frictionless motion, changes in these variables between two time steps along a trajectory are attributed to diabatic processes, such as cloud formation, radiation and friction. Following the method of Pfahl et al. (2015) and Steinfeld and Pfahl (2019), the effect of LH is quantified by the percentage of blocking trajectories with a maximum heating (Lagrangian change of $\theta$) of $\Delta\theta > 2$ K during the three days prior to reaching the blocking region (in the following denoted as LH contribution). Trajectories with $\Delta\theta < 2$ K, which also comprises air masses that experience net cooling along the flow, are classified as quasi-adiabatic trajectories. To define WCB trajectories, a slightly weaker ascent criterion of 500 hPa in 48 h is applied than in Madonna et al. (2014b) with 600 hPa in 48 h.

- **PV advection:** Considered as an indirect diabatic effect of LH (Davis et al., 1993), the effect of the divergent outflow on the structure and development of blocking is evaluated here by calculating the PV advection by the divergent (irrotational) component ($\mathbf{v}_\chi \cdot \nabla \mathrm{PV}$) of the full wind following Riemer et al. (2008) and Archambault et al. (2013). The divergent wind is obtained via Helmholtz partitioning, using a successive overrelaxation method. In addition, the role of the rotational (non-divergent) wind component ($\mathbf{v}_\psi$) is investigated, which highlights the contribution of the balanced flow associated with the upper-level PV distribution. PV advection by the divergent and rotational wind is averaged vertically between 500 - 150 hPa.

One limitation of this methodology is that the trajectories follow the resolved large-scale wind and do not capture fast convective motions. This might introduce an underestimation of the contribution of LH from convection for the upper-level flow evolution (Oertel et al., 2020).

## 2.4 Selection of the cases

Atmospheric blocking covers a variety of flow patterns, including $\Omega$-shaped or high-over-low dipole blocks, which can occur all year round in different regions (Woollings et al., 2018). There is a large case-to-case and spatial variability of the LH contribution to blocking, ranging between 35 - 55 % for the majority of observed blocking cases in the global ERA-Interim climatology (Steinfeld and Pfahl, 2019). To cover part of this variability we perform sensitivity experiments for five different blocking events, which develop under different environmental conditions (different seasons, geographical locations and LH contribution), as summarized in Table 1. Blocks are selected from the main blocking regions over the North Atlantic and North Pacific, but also from a secondary region over Russia. Some of those blocks are associated with extreme weather events: the 2010 summer heat wave in western Russia, the devastating wildfires in Alberta, Canada in May 2016 and the cold spell in Europe in February 2018 (dubbed the "Beast from the East").

One of these cases, Thor (onset and maintenance) in the year 2016, is used hereafter to introduce our method. Therefore, its evolution is described in detail in the following section.

## 3 Case Study: Block Thor

Block "Thor" occurred over the North Atlantic and Europe in the period 2–19 October 2016, during the North Atlantic Waveguide and Downstream Impact Experiment (NAWDEX; Schäfler et al., 2018). The onset of Thor was associated with forecast uncertainty, in particular the predictability of the upstream cyclone and its diabatic outflow was low (Maddison et al., 2019, 2020).

The complex interaction between an upstream cyclone, latent heating and the upper-level flow during the onset of Thor on 2 October 2016 is qualitatively illustrated in Fig. 1a, with cloud top pressure from MSG satellite measurements overlaid over the IFS CNTRL simulation after 2 days lead time. An intensifying North Atlantic cyclone [named Stalactite cyclone in Schäfler et al. (2018)] is associated with increased diabatic heating in the WCB, indicated by an elongated band of high-reaching clouds, and an upper-level trough that wraps cyclonically around the surface low (labelled T1). The outflow of this ascending

and cloud-producing airstream concurs with a pronounced poleward displacement of the upper-level PV contours (labelled R1). This ridge building marks the onset of block Thor. Ten days later on 11 October 2016, Fig. 1b shows the maintenance phase of block Thor. At this point in time, Thor (R2) is characterized by a region of low upper-level PV, high surface pressure and subsidence with low or no clouds over Scandinavia. Further upstream over the North Atlantic, two cyclonic systems (T3 and T4) are associated with strong cloud activity and ridge amplification (R3 and R4).

As the block persisted for more than 2 weeks, two simulations are performed here capturing the onset (Thor onset: 30 Sep–10 Oct) and the maintenance/decay (Thor maintenance: 10 Oct–20 Oct) phases. Note that only the second period was named "Thor" in Schäfler et al. (2018), and the first period was referred to as Scandinavian blocking. However, from a "PV-anomaly" perspective, the entire episode can be described as one persistent blocking event.

### 3.1 Synoptic overview

The life cycle of Thor is characterized by a succession of multiple upstream triggers over the North Atlantic, i.e. synoptic-scale baroclinic waves, their dynamic interaction with the jet stream and the subsequent formation and maintenance of a downstream blocking anticyclone. Fig. 2 shows the temporal evolution of the LH contribution, mean diabatic heating along blocking air masses, blocking intensity and spatial extent for Thor in the two CNTRL simulations (onset and maintenance), and Figs. 3a,c and 4a,c display aspects of the block's evolution at upper levels (upper-level PV and 500 hPa geopotential height (Z500) giving an indication of the large-scale upper-level flow with the jet stream following the band of enhanced PV/Z500 gradient) and lower levels (sea level pressure (SLP) and diabatic heating). On the basis of the PV-anomaly index, block Thor is tracked from 2–9 October (onset simulation) and 11–19 October (maintenance simulation).

Thor shows typically observed blocking characteristics (e.g., Dole, 1986), such as the rapid onset (fast increase in intensity and spatial extent, Fig. 2) on time scales consistent with synoptic-scale phenomena (2–4 October) and the fluctuation in intensity and size during the blocking lifetime (mature phase: 5–17 October) until its decay (19 October). The episodic nature of the LH contribution and the mean diabatic heating highlight the importance of LH changes throughout the life cycle, alternating between times when either moist-diabatic (heating) processes or quasi-adiabatic (cooling mostly due to long-wave radiation) processes dominate: the LH contribution is generally largest during onset (70 %) and then declines to the lowest value (almost 0 %) when the block decays. However, there are multiple bursts of LH (local maxima of LH) during the life cycle, which are followed by fluctuations in intensity and size. The block exhibits its most rapid amplification during such LH bursts, suggesting that there is a linkage between moist-diabatic processes and the development of the block. Averaged over the entire lifetime (onset and maintenance), Thor has a LH contribution of 41 %, that is almost half of the blocking air masses have been diabatically heated by more than 2 K.

This episodic nature of LH emphasizes that a series of upstream transient cyclones, rather than a single primary cyclone, contribute to block formation and maintenance. In this case, the upstream triggers (in total 5) include a rapidly intensifying cyclone [denoted as Stalactite cyclone in Schäfler et al. (2018) and Maddison et al. (2019)] ahead of an upper-level PV trough (labeled T1 in Fig. 3a), which initiates downstream ridge building R1 and the subsequent onset of the block on 2 October 2016. This is followed by a rapidly propagating surface cyclone T2 from the southwest along an intense baroclinic zone with strong

poleward transport of low-PV air in secondary ridge R2, which further intensifies and expands the initial blocking ridge formed by R1 (outlined by the violet contour) and finally leads to anticyclonic wave breaking and the establishment of a stationary dipole block over Europe (Fig. 3c), resembling the classic dipole blocking structure with a negative PV anomaly to the north of a positive PV anomaly (or a positive geopotential anomaly north of a negative geopotential anomaly) described by Berggren et al. (1949) and Rex (1950). Maximum intensity of the simulated blocking in terms of upper-level negative PV anomaly and spatial extent occurs around 8 October (8 days into the Thor onset simulation, see Fig. 2). During this mature phase, which extends into the maintenance simulation (Fig. 4a), the block R2 stays well established and stationary over Scandinavia for the next few days, as the dipolar configuration over Europe generates an easterly flow at the latitude of the jet (60°N), which counters the advection by the background westerly flow. This is also the time when absolute reversal blocking indices (e.g., the index described by Scherrer et al., 2006) identify the block (not shown). At this point in time, the block is associated with a barotropic signature with a surface high pressure system and a tropospheric-deep anticyclonic flow, splitting the upper-level westerly flow into northern and southern branches, as indicated by the Z500 contours in Fig. 4a. In this split/diffluent flow region on the western side of Thor, a meridionally-elongated PV filament T3 develops and is associated with cloud formation and poleward transport of low-PV air along its eastern flank in ridge R3. T3 and R3 are stretched meridionally between block Thor (R2) and a quickly amplifying ridge R4 to the west and the air masses in R3 are absorbed into the blocking anticyclone (Fig. 4a). R4, associated with intense cloud formation in an ex-tropical cyclone T4 over the east coast of North America (see again Fig. 1b), extends rapidly and replaces R2 and R3, thus maintaining a strong and large negative PV anomaly over Northern Europe, contributing to the blocks' persistence (not shown). There is a last absorption of low-PV air in ridge R5 before Thor finally decays (Fig. 4c). For this particular event, lysis is a comparatively slow process and is characterized by a synchronous decrease in the intensity and spatial extent while the block slowly moves southeastward (not shown).

The trajectory analysis in Fig. 5 illustrates the origins and flow history of low-PV air in the blocking anticyclone. Shown are backward trajectories emanating from the block during onset (Fig. 5a) and maintenance (Fig. 5c). It reveals two distinct types of airstreams: The first type (marked with black triangles at day -3) consists of upper-level trajectories that either (i) originate from the west and flow quasi-horizontally (and quasi-adiabatically, i.e. weak radiative cooling and small diabatic PV modification, Fig. 5e) along the upper-level jet (around the upstream trough) into the block (most evident during onset) or (ii) are already located in the blocking region at day -3 and recirculate anticyclonically within the block (evident during maintenance). The second type (black circles at day -3) consists of trajectories that ascend rapidly from low levels ($> 800\,\mathrm{hPa}$) to higher levels ($< 500\,\mathrm{hPa}$) ahead of surface cyclones over the North Atlantic. These trajectories are heated by $\sim 10\,\mathrm{K}$ in the median, experience net diabatic reduction of PV and reach the upper troposphere with very low PV values ($< 0.3\,\mathrm{pvu}$, Fig. 5e), which corresponds to substantial negative PV anomalies (of roughly -1 pvu in the median). Each scheme (cloud microphysics and convection) contributes about $5\,\mathrm{K}$ to the total diabatic heating along these ascending trajectories (not shown), pointing towards the importance of convective processes for the generation of the negative PV anomaly (cf. Rodwell et al., 2013; Oertel et al., 2020). This cross-isentropic ascent occurs primarily on the western flank of the block in regions of strong cloud activity (see again Fig. 1), intense latent heat release and upper-level divergent outflow (Figs. 3a,c and 4a,c). 15 % of these ascending trajectories fulfil the WCB criterion of 500 hPa ascent in 48 hours and are heated by more than 20 K in three days. The median

evolution of $\theta$ and PV (Fig. 5e) along these two types of trajectories shows that the heated trajectories ($\Delta\theta > 2\,\mathrm{K}$, yellow line) typically reach higher (isentropic) altitudes with lower PV values compared to quasi-adiabatic trajectories ($\Delta\theta < 2\,\mathrm{K}$, blue line), which underlines the importance of LH to generate intense upper-level anticyclonic PV anomalies.

## 3.2 Synoptic overview for the other cases

Figures showing the synoptic evolution at 2 and 6 days lead time for the other three cases (Cold spell, Canada and Russia) can be found in the supplement (Figs. S1, S2 and S3).

Consistent with the evolution of Thor, the other blocking cases are initiated by and interact with upstream extratropical cyclones. For the Cold spell case (Fig. S2), two North Atlantic upstream cyclones are present during onset (day 2, labelled T1) and the second intensification phase (day 6, labelled T2). The Canada case (Fig. S1) is only affected by one North Pacific upstream cyclone during onset (day 2, labelled T1), but this cyclone moves slowly and influences the block for the next 4 days. The block in the Russia case (Fig. S3) is initiated by a North Atlantic cyclone during day 2 (labelled T1). It then propagates further eastward and reaches its maximum amplitude over Western Russia at day 6.

Trajectory analysis for the cases Russia, Canada and Cold spell shows a flow behavior similar to Thor, with air masses that either (i) flow quasi-adiabatically or (ii) ascend cross-isentropically into the blocking region ahead of a cyclone (not shown), with the strongest LH contribution during onset. Case Canada has a mean LH contribution of 52 %, and case Russia and Cold spell have a mean LH contribution of 42 % and 38 %, respectively (Table 1).

These cases are typical of block formation after explosive cyclogenesis (e.g., Colucci, 1985; Lupo and Smith, 1995; Maddison et al., 2019) with rapid ridge amplification of transient waves into a large-amplitude block (Altenhoff et al., 2008), and reinforcement by mid-latitude eddies propagating into the strong deformation field on the western flank of the block (Shutts, 1983; Colucci, 2001), resulting in large-amplitude upstream troughs and ridges and the subsequent replacement and/or absorption of 'fresh' low PV air by the block (Yamazaki and Itoh, 2012; Luo et al., 2014; Steinfeld and Pfahl, 2019). While blocking patterns appear stationary, the upper-level flow is hence highly dynamic with old anticyclonic air masses being replaced by new ones.

## 4 Sensitivity experiments

In the synoptic evolution of Thor and the other cases, we observed the presence of upstream LH during the life cycle of blocking. However, the extent to which the formation and maintenance of blocking was forced by LH remains unclear. As noted earlier the advection of low PV into the core of a block often alternates between moist-diabatic injection of air from the lower troposphere and quasi-adiabatic advection of upper-level air. To isolate and assess the impact of LH on blocking, in the following we compare the NOLH simulations without LH to the CNTRL simulations with LH.

We first provide a synoptic comparison between CNTRL and NOLH for Thor onset and maintenance, which helps to illustrate the sensitivity experiments before discussing all five cases as well as the case-to-case variability.

## 4.1 Thor: Synoptic differences with and without LH

Backward trajectories from Thor identify the North Atlantic storm track as the relevant diabatic heating region (see again Fig. 5a,c). Across much of the basin the heating (gray contours in Figs. 3a,c and 4a,c) occurs in the warm sector of cyclones. Therefore, the NOLH box is placed over [60°W - 0°, 35°N - 65°N], covering the entire North Atlantic basin, as indicated by the black box in the right panels of Figs. 3 and 4.

Figure 5b,d shows that no strongly ascending air masses contribute to the ridge amplification in the NOLH simulations. During blocking onset (Fig. 5b), mostly quasi-adiabatic and quasi-horizontal flow is associated with Thor. In the maintenance simulation, which is initialized with a mature dipole block (Fig. 5d), the block is associated with quasi-adiabatic upper-level trajectories that recirculate anticyclonically within the blocking anticyclone, without the ascending airstreams linked to troughs T3 and T4 in the CNTRL simulation.

Turning off LH over the North Atlantic thus effectively reduces cross-isentropic transport, and reduces the average LH contribution from 41% (CNTRL) to 16.5 % (NOLH). Note that the remaining heated trajectories in NOLH experience considerably less heating (median of ∼2 K compared to ∼10 K in CNTRL, Fig. 5e,f), but still with PV reduction along the flow, most likely due to ice microphysical process (e.g., depositional growth of snow and ice, see Joos and Wernli, 2012) at higher altitudes above the NOLH box (cf. method section). Overall, the non-heated quasi-adiabatic trajectories in NOLH show a similar behavior as in the CNTRL simulations.

Given the changes in LH contribution and diabatic heating along the blocking trajectories, we now focus on the impact of LH on the upper-level synoptic-scale flow evolution of Thor. Figures 3 and 4 compare upper-level PV, Z500, upper-level divergent wind, SLP and lower-level cloud-diabatic heating from the NOLH simulations to the corresponding results from the CNTRL simulations. Note that the differences between CNTRL and NOLH are initially weak (after 2 days in the Thor onset and Thor maintenance simulations), but become more pronounced with lead time. Nevertheless, the early evolution highlights the critical phase when the two simulations start to deviate.

### 4.1.1 Thor onset

After 2 days, shortly before the incipient block in the CNTRL simulation is identified, remarkable differences in the upper-level PV and Z500 between the CNTRL (Fig. 3a) and NOLH (Fig. 3b) simulations emerge in the region of ridge R1, with the largest differences in the dynamically active regions associated with the latent heat release and divergent outflow of the heated trajectories. A trough-ridge pattern evolves also in NOLH due to dry baroclinic development of T1, but, in the absence of LH, the amplitudes of the upper-level PV and Z500 ridges and troughs, as well as the intensity of the upstream cyclone (see SLP contours in Fig. 3a,b) are clearly reduced. This leads to a delayed onset of the block in NOLH compared to CNTRL by one day.

Differences in the upper-level divergent wind are substantial, indicating that diabatic heating significantly enhances the vertical motion and divergent outflow. Moist dynamics account for roughly two thirds of the divergent outflow, which exceeds $10\,\mathrm{m\,s^{-1}}$ on the western flank of R1 in CNTRL compared to $< 3\,\mathrm{m\,s^{-1}}$ in NOLH. In the CNTRL simulation, a comma-shaped

diabatic heating pattern is co-located with the divergent outflow aloft, which compares favorably with the cloud patterns in the satellite observations (Fig. 1a) and the ascending heated trajectories (Fig. 5a). The divergent wind above the cloud-diabatic heating maximum in CNTRL aids the westward expansion of ridge R1 through the westward advection of air masses with low PV, shifting the tropopause in the same direction and considerably strengthening the PV gradient (see details on PV advection in Section 4.2.2). The cyclonic wrap up of high- and low-PV in the upstream trough T1 does not occur in NOLH, suggesting that this cyclonic wave breaking and the horizontal rearrangement of upper-level PV depends essentially on intense LH.

Further into the model integration on day 6 (Fig. 3c,d), the differences between CNTRL and NOLH are considerably more pronounced and it is clear that the large-scale flow develops substantially differently without LH. With the contribution of LH in CNTRL, the secondary ridge R2 rapidly amplifies and low-PV air is transported a long way poleward, causing (i) a south-westward extension of the initial blocking region and (ii) a reinforcement of the anticyclonic anomaly formed by ridge R1 over Scandinavia (Fig. 3c). The upper-level flow splits over central Europe with an accelerated southwest - northeast tilted northern branch (jet stream), evident from the Z500 contours in Fig. 3c. When LH is turned off, however, the ascent and outflow are significantly reduced and ridge R2 does not amplify (Fig. 3d). This is consistent with the position of T2 being too far south. Instead, R2 is deflected eastward by the westerly winds. As a consequence, the low-PV region of R1 is cut off from the tropospheric reservoir and a zonally oriented jet stream establishes over western Europe. Without LH, PV values inside R1 are higher, i.e. the upper-level negative PV anomaly is weaker, resulting in a less pronounced anticyclonic flow over Scandinavia, as also evident from the Z500 contours. The upper-level synoptic features in NOLH are displaced further downstream, where the flow still splits with a weaker northern branch compared to CNTRL.

### 4.1.2 Thor maintenance

To better understand the role of LH for the persistence of blocking, we now focus on the Thor maintenance simulation. Both CNTRL and NOLH simulations start with a well established dipole block over Europe and a large-scale diffluent flow field upstream (visible in the Z500 contours), where a large region with low upper-level PV values covers most of Scandinavia on day 2 (R2 in Fig. 4a,b). However, first pronounced differences in the divergent outflow strength and the upper-level PV structure occur in the region of upstream ridge R4 to the east of trough T4. In the absence of LH, ridge R4 and consequently the PV streamer T3 are not as strongly extended in the meridional direction as they are in CNTRL, despite being subject to a strong diffluent flow, suggesting that the (dry) eddy straining mechanism (Shutts, 1983) does not fully explain the amplification of the incoming upstream waves. As a consequence, R4 in NOLH does not replace the initial negative PV anomaly R2 over Scandinavia (cf. Fig. 4c,d). Without the diabatic contribution of 'fresh' low-PV air, and facilitated by the radiative decay (cooling and net PV increase along upper-level trajectories) of the remaining air masses recirculating inside the block (Fig. 5d,f), Thor weakens in the NOLH simulation and is no longer captured by the blocking index on 15 October (day 5). In contrast, the CNTRL block persists for another 4 days, also due to the additional absorption of anticyclonic air masses in R5 on day 6 (Fig. 4c,d).

### 4.1.3 Non-linear effects of latent heating

In order to exemplify the non-linearity of the relationship between LH and blocking, Fig. 6 shows the 2 pvu tropopause at day 3 and day 6 of Thor onset with and without LH, and also with reduced LH ($\alpha = 0.5$) and increased LH ($\alpha = 1.5$). The evolution of the tropopause shows a crucial sensitivity to changes in LH with a non-monotonic behaviour of blocking to LH. Note that the modifications of LH first become apparent in the region of the NOLH box over the North Atlantic and Europe and only spread out at longer lead times. During the onset phase (day 3, Fig. 6a), the ridge has a larger amplitude and extends further to the west over Greenland with increasing LH, with cyclonic wrap up of high- and low-PV in the upstream trough most evident in the simulation with enhanced LH ($\alpha = 1.5$, red contour). Consequently, also the downstream trough is more amplified and narrows into a PV streamer in the simulations with unchanged (CNTRL, yellow contour) and enhanced LH. The northward and westward amplification of the ridge is weaker in the simulations with reduced (green contour) and removed (NOLH, blue contour) LH. During the mature phase (day 6, Fig. 6b), LH ($\alpha = 1$ and $\alpha = 1.5$) leads to anticyclonic wave breaking and the formation of a stationary dipolar flow pattern over Europe with low-PV to the north of a cut-off high-PV anomaly. In addition, the eastward propagation and zonal extent of the upstream trough is slowed down, an effect of LH also observed by Ahmadi-Givi et al. (2004). When LH is reduced or switched off, the ascent and outflow are reduced (not shown), the ridge does not amplify as strongly and, in the absence of wave breaking, blocking is not initiated. Instead, the low-PV region is cut off from the tropospheric reservoir, surrounded by high-PV stratospheric air and located further north above Svalbard.

The comparison of block Thor with and without LH reveals some interesting differences and helps understanding the causal relationship between LH and blocking during the initiation and maintenance/decay phases. This example illustrates how LH in ascending airstreams embedded in upstream cyclones can play a crucial role in the initiation, but also in the maintenance of blocking, contributing to a more rapid development and longer lifetime of the block. This strong sensitivity of block development to changes in upstream latent heating further suggests that forecast uncertainty during blocking can arise from diabatic heating from parametrized processes (e.g., Grams et al., 2018; Maddison et al., 2019). Moist-diabatic processes provide further flow amplification in addition to dry-dynamical forcing, and repeated diabatic injection of low PV can extend the lifetime of a block and diminish the tendency for dissipation.

## 4.2 Set of blocks: Differences with and without LH

To evaluate the sensitivity experiments in a more robust and systematic way, we analyze a set of 5 historical blocks in total over different regions and in different seasons (see again Table 1).

### 4.2.1 Differences in upper-level PV structure

Figure 7 shows the differences in the upper-level PV and upper-level divergent wind between the NOLH and CNTRL simulations (CNTRL - NOLH) during onset at day 3 for the five blocking cases. While synoptic Figures above (Figs. 3,4 and S1, S2 and S3) show that CNTRL and NOLH simulations start to deviate at day 2, by day 3 there are distinct differences in the upper-level PV field.

In all cases, the dynamical tropopause (2 pvu contour) is displaced much farther to the pole and west in the regions associated with the divergent outflow in the CNTRL simulations, along with pronounced differences in the upper-level PV between the CNTRL and NOLH. The absence of LH results in higher PV and thus in weaker anticyclonic anomalies in NOLH, which is reflected in negative PV differences of more than -1 pvu between CNTRL and NOLH, reaching -3 pvu in cases Thor onset and maintenance, Cold spell and Russia. At this time, the center of mass of the tracked blocks in the NOLH simulations corresponds well with the blocking centre in the CNTRL simulations (crosses and pluses in Fig. 7).

Despite the difference in the synoptic environment between the five cases, it becomes evident that in each case strong LH embedded in the upstream cyclone substantially contributes to this initial ridge amplification and the onset of the blocks. The most pronounced PV differences are co-located with the tropopause, i.e., the region of enhanced PV gradient, which has important implications for the propagation of Rossby waves in the upper troposphere (Schwierz et al., 2004b; Martius et al., 2010). The more pronounced ridge also results in a more amplified downstream flow pattern in CNTRL, with the downstream trough penetrating further equatorward in all cases.

Differences in the upper-level divergent wind between CNTRL and NOLH are substantial in all cases (more than $5\,\mathrm{m\,s^{-1}}$, see wind vectors in Fig. 7). Given that the total upper-level divergent wind in the CNTRL simulations is generally less than $10\,\mathrm{m\,s^{-1}}$ near the western flank of the ridges (see wind vectors in left panels of Figs. 3,4 and S1, S2 and S3), these differences are considerable and it is clear that strong vertical motion (not shown) and upper-level divergence arise from LH. The diabatically enhanced divergent outflow tends to facilitate the westward and poleward expansion of the ridge by advecting low PV in these directions. In addition to the diabatic injection of low PV air from the lower troposphere in ascending airstreams (shown as an example for case Thor in Fig. 5e,f), the divergent outflow contributes to the pronounced upper-level PV differences along the western flank of the ridges through this effect (see subsection 4.2.2).

A few days later during the mature phase (6 days into the simulations), Fig. 8 shows substantial differences in the upper-level PV and upper-level rotational wind between the CNTRL and NOLH simulations. The initial PV differences confined to the north-western flank of the ridges during onset, i.e. early phase of the simulations, have amplified and propagated up- and downstream, leading to distinctively different evolutions of the upper-level flow with strongly displaced ridges and troughs and marked differences in the upper-level PV pattern. In all cases, the intensity and spatial extent of the blocks are reduced in NOLH, which is reflected in negative PV differences between CNTRL and NOLH. The largest differences ($\Delta$PV < -3 pvu) are found inside the blocking region, especially in the core (close to the center of mass in Thor onset, Thor maintenance, Canada and Russia) and around the flanks of the block (Cold spell). Positive and negative upper-level PV differences are also found in the upstream and downstream troughs and ridges, indicating a shift in location. The diabatic intensification of the blocks in CNTRL goes along with an amplified upper-level anticyclonic circulation (see wind vectors in Fig. 8). The differences in the upper-level rotational wind clearly reveal the intensified anticyclonic flow associated with the intense negative PV anomalies of the CNTRL simulations, especially on the flanks around the negative PV differences with substantial wind speed differences of up to $40\,\mathrm{m\,s^{-1}}$ between CNTRL and NOLH.

In the following, we take a closer look at the individual cases. In Thor onset (Fig. 8a), negative PV differences inside the block and positive differences south of it indicate the anticyclonic wrap up of low- over high-PV air and the formation of a

dipole block with easterly winds in CNTRL, while in NOLH the negative PV anomaly is detached further north above Svalbard as a tropospheric cut-off.

In Thor maintenance (Fig. 8b), the block is still present in CNTRL while it is already too weak to be detected in NOLH. The poleward elongation of the CNTRL block is reflected in the negative PV difference ($\Delta$PV up to -4 pvu) with an anticyclonic flow centered over Iceland. In NOLH, the decaying blocking ridge over Europe and the cut-off PV anomaly east of Greenland do not merge (see discussion above).

For the case Canada (Fig. 8c), the omega-shaped structure of the block with tilted upstream and downstream troughs is not reproduced without LH, and the NOLH block develops as an open ridge embedded in a Rossby wave with a weak anticyclonic circulation over western Canada.

In the case of Russia (Fig. 8e), the initial PV differences over western Europe have propagated eastward and reach values of -5 pvu further downstream over western Russia at day 6, with a strong anticyclonic flow only present when LH is included.

In contrast to the other cases, the PV values inside the block's core are similar in CNTRL and NOLH for the Cold spell case (Fig. 8d). Largest negative PV differences are found along the edge of the block, i.e. the block is smaller in spatial extent in NOLH, and further south over the Azores, where the NOLH block detaches from the tropospheric reservoir.

Interestingly, a common feature in several NOLH simulations (Thor onset, Thor maintenance, Cold spell and Russia) is the formation of a low-PV anomaly that is cut off from its tropospheric source and surrounded by high-PV stratospheric air (closed dashed contours in Fig. 8a,b,d,e). These cut-off anomalies are formed when the jet stream is retreating back to a more zonal flow. In contrast to the CNTRL simulations, they are not accompanied by a cyclonic anomaly to the south, and therefore do not constitute a stationary dipolar flow pattern that generates stronger easterlies at the primary latitude of the jet. The typical inverse-S shape of the 2 pvu contour during overturning Rossby waves, which is used to describe blocking in association with wave breaking (e.g., Pelly and Hoskins, 2003) is only simulated with the inclusion of LH. The formation of such cut-off blocks in synoptic situations with reduced LH contribution is also in agreement with the climatological composites in Steinfeld and Pfahl (2019). This again highlights the role of LH in effectively displacing the jet stream far to the north and promoting persistent anticyclonic Rossby wave breaking.

### 4.2.2 Differences in PV advection by the divergent outflow

For a quantitative analysis of the indirect effect of LH on upper-level PV (Davis et al., 1993; Stoelinga, 1996), Fig. 9 shows the difference in divergent wind ($\mathbf{v}_\chi$) and associated PV advection by the divergent wind ($\mathbf{v}_\chi \cdot \nabla$PV) between CNTRL and NOLH simulation for the five cases during an early phase of the simulations, i.e., during the most intense ridge amplification at 3 days lead time. The strong enhancement in divergent outflow aloft by LH is accompanied by a stronger negative upper-level PV advection on the north-western flank of the blocking ridge in all cases, locally with differences of -0.3 pvu h$^{-1}$ between CNTRL and NOLH. Given that the upper-level PV advection by the divergent wind in the CNTRL simulations reaches absolute minimum values of -0.4 pvu h$^{-1}$ at this time (not shown), these differences are considerable. Thus, the negative PV advection on the western flank is almost absent in NOLH.

It is important to note that, while the upper-level divergent wind is generally one order of magnitude smaller than the upper-level rotational wind, the PV advection by the two wind components is of much more similar magnitude since the divergent wind is typically parallel to the upper-level PV gradient (Steinfeld and Pfahl, 2019). For all cases, the negative PV advection by the divergent wind counteracts the positive PV advection by the rotational wind on the north-western flank during onset, resulting in a reduced positive (for cases Thor maintenance and Canada) or even in a net negative (for cases Thor onset, Cold

spell and Russia) PV advection by the total wind (($\mathbf{v}_\chi + \mathbf{v}_\psi$) · ∇PV, not shown). This negative PV advection by the divergent wind on the western flank contributes to the initial negative PV differences seen in Fig. 7 and therefore contributes to the westward extension and quasi-stationary (slower eastward progression) behavior of blocking (Mullen, 1987; Steinfeld and Pfahl, 2019), an effect of LH on upper-level waves also observed in the sensitivity studies by Davis et al. (1993) and Stoelinga (1996). Since forecast uncertainties during blocking onset often manifest on the western flank of the ridge (Matsueda, 2011;

Quandt et al., 2018), we hypothesize that this is associated with the divergent outflow.

### 4.2.3   Differences in blocking characteristics and case-to-case variability

Figure 10 shows a quantitative comparison of the temporal evolution of blocking characteristics (LH contribution, mean diabatic heating along all blocking trajectories, intensity and spatial extent) obtained from the CNTRL (solid lines) and NOLH (dashed lines) as a function of simulation lead time. Note that the individual curves start as soon as a block is identified with

the PV-anomaly index (see section 2) in the corresponding simulation. Characteristics based on 3-day backward trajectories (LH contribution and diabatic heating) can only be obtained after at least 3 days of model integration time.

The episodic nature of LH contribution and diabatic heating (Fig. 10a,b) during the blocking life cycle in the different CNTRL simulations (solid lines) is associated with the passage of synoptic cyclones and the associated cross-isentropic transport of low-PV air in WCBs. LH bursts (local maxima of LH contribution and diabatic heating) typically indicate the time

of strongest interaction between the block and the approaching upstream cyclones (see also Steinfeld and Pfahl, 2019). The periods between such LH bursts are dominated by a median cooling of -3 to -4 K and predominantly quasi-horizontal transport of near-tropopause air masses (see again quasi-adiabatic trajectories in Fig. 5). The relative importance of LH varies strongly during the lifetime of the CNTRL blocks and from system to system, but is generally largest during onset (LH contribution of around 60 %) and then declines to the lowest value when the blocks decay, consistent with the climatological analysis in

Steinfeld and Pfahl (2019). In contrast to the cases with multiple LH bursts (Thor onset and Cold spell) or with a prolonged strong LH contribution from one slowly moving upstream cyclone (Canada), cases Thor maintenance (blue solid line) and Russia (violet solid line) experience strong diabatic heating only during the early phase in CNTRL, and after day 5 mostly quasi-adiabatic advection of low PV, i.e. cooling along upper-level air masses, dominates the evolution of the blocks.

Considering all the blocks in the CNTRL simulations, 43 % of their air masses experience heating of more than 2 K in 3 days,

and the median heating along the heated trajectories is 11 K, with a wide range of $\Delta\theta$ up to 45 K for individual trajectories (not shown). 10 % of the heated trajectories are classified as WCBs. In the NOLH simulations, the LH contribution is not entirely removed (cf. method section), but reduced to 15 % and a net diabatic cooling of -3 to -4 K in the median dominates the entire

evolution of the blocks (dashed lines in Fig. 10b). The remaining 15 % of heated trajectories experience only weak heating of 3 K in the median (not shown), and only 2 % fulfill the WCB criterion.

Comparing the evolution of block intensity and spatial extent between CNTRL and NOLH in Fig. 10c,d shows that LH leads to more intense and larger blocks (in all cases) with an extended lifetime (Thor maintenance). In the CNTRL simulations, blocking ridges intensify more rapidly during their early growth phase (days 1 - 4) and upper-level PV anomalies are thus stronger and spatially more extended compared to their counterparts without LH. However, the experiments indicate a large case-to-case variability with respect to the sensitivity of the block to LH. Without LH, the Thor onset (red lines) and Cold spell (yellow lines) blocks develop later with a delay of about 1 and 4 days respectively, because the first ridge amplification is too weak. Only, later, when a secondary upstream cyclone is approaching (at days 4 - 5 for Thor onset and at day 6 for Cold spell, indicated by the second maxima in the LH contribution/diabatic heating for the CNTRL simulations), does the anomaly in NOLH become stronger, even reaching similar blocking intensities to CNTRL (around day 7), though smaller in extent. Likewise, the Russia block (violet lines) has a delayed onset and a slower amplification without LH, but has a similar intensity at day 7 as the CNTRL block, which begins to decay after attaining peak intensity around day 4. The Canada block has its onset at the same time in both CNTRL and NOLH simulations (green lines), however the ridge does not further amplify in NOLH and differences in intensity and spatial extent between NOLH and CNTRL increase with model integration time. The Thor maintenance block (blue lines), which starts as an intense and large-scale anticyclonic anomaly in both CNTRL and NOLH simulations, experiences a quick reduction in amplitude without LH, and dissipates 4 days earlier than the CNTRL block.

Since the characteristics of the block can develop differently, it is difficult to quantify which event is most sensitive to changes in upstream LH. The normalized differences in peak intensity and spatial extent between the NOLH simulations and each corresponding CNTRL simulation are shown together with the LH contribution from the CNTRL simulations (see again Table 1) in Fig. 11. Since for Thor maintenance both simulations start with a mature block, differences are shown for 5 days lead time (before the NOLH block is too weak to be identified by the blocking index). Blocks with a small sensitivity to changes in upstream LH will have values close to zero (i.e. no large differences between NOLH block and CNTRL block). A value of -0.5 represents a reduction by a factor of 2. This figure shows again that all NOLH simulations exhibit a reduction in peak intensity and spatial extent. The reduction is largest for Canada (around -0.3 for intensity and -0.7 for extent) and smallest for Thor onset, Cold spell and Russia (around -0.1 for intensity and -0.4 for extent). The Canada block also has a large LH contribution in its CNTRL simulation (52 %). However, Thor onset with a similar LH contribution (47 %) shows less sensitivity and a weaker reduction in intensity/spatial extent, the latter being more similar to Cold spell with a smaller LH contribution of 38 %. In addition, the Thor onset simulations with reduced LH ($\alpha$ = 0.5) and enhanced LH ($\alpha$ = 1.5) are shown as open red circles, highlighting that the sensitivity of blocking in the Thor onset case is not linear with respect to changes in LH. It shows that an increase in LH has an even greater effect on spatial extent than on intensity, as blocking area increases by a value of 0.7 (by a factor 3) for $\alpha$ = 1.5.

The effect of LH on blocking intensity and extent appears to depend not only on the LH contribution, but also on other environmental features such as the phase of the blocking life cycle, the number and strength of LH bursts/upstream cyclones, and the state of the background flow. During the early growth phase with an initially zonal and intense upper-level jet stream,

cloud diabatic heating intensifies the upstream cyclone and facilitates a faster growth of the incipient ridge. Since case Canada interacts with only one upstream cyclone with particularly large and prolonged LH contribution, the generation of the upper-level PV anomaly strongly depends on LH and its removal has profound effects on the upper-level flow evolution (omega block in CNTRL vs open ridge in NOLH, see again Fig. 8c). However, during the mature phase (after 4 days lead time) when the large-scale flow is already in an amplified state, the ridges in Thor onset and Cold spell interact with downstream propagating waves and amplify in NOLH, and thus they appear less sensitive to changes in LH. The presence of an amplified ridge with a large-scale upper-level diffluent flow is known to provide a favorable environment for blocking initiation and maintenance (Colucci, 1985; Pelly and Hoskins, 2003), which supports the meridional amplification of the upstream waves [eddy straining mechanism, Shutts (1983)] and the (isentropic) poleward transport of air with low PV (Yamazaki and Itoh, 2012; Steinfeld and Pfahl, 2019), and the block can thus also develop in the absence of intense LH, though smaller in extent. Dry-dynamical forcing alone, however, is not able to maintain the Thor block in the absence of LH, and after 5 days lead time in the Thor maintenance simulation the blocked region is reduced by -0.7 (approx. by a factor 3). In contrast to the other cases, the blocking ridge in case Russia propagates downstream over Russia and further away from the storm track region over the North Atlantic ocean basin in both simulations (see Fig. 8d), and therefore away from the influence of direct diabatic injection of low-PV air. Thus, the evolution of the Russia block after its onset is mostly governed by quasi-adiabatic dynamics in both CNTRL and NOLH simulations (violet lines in Fig. 10a,b after day 5). It may be related to downstream propagating wave trains emanating from the North Atlantic that interact with topographically-forced planetary waves (see Nakamura et al., 1997; Luo et al., 2016). Climatologically, blocks over Russia typically form with small LH contribution (below 20 %, see Fig. 5 in Steinfeld and Pfahl (2019)) which may explain the small sensitivity of the Russia case. However, despite similar sensitivities of blocking intensities in the Thor onset, Cold spell, Russia and Thor maintenance cases, there is still a big difference in the large-scale flow evolution between the simulations (dipole block in CNTRL vs cut-off low-PV anomaly in NOLH, see again Fig. 8a,b,d,e). Generally, the sensitivity of blocking intensity is smaller than the sensitivity of spatial extent, suggesting that comparing blocks based on their intensity only might hide some of the synoptic differences.

Despite the strong case-to-case variability in the LH contribution and in the sensitivity of the blocks to changes in LH, the experiments demonstrate that LH can have a profound effect on blocking intensity, spatial extent and lifetime. As mentioned above, in all cases, except for Cold spell, the tracked negative PV anomalies are not classified as blocking in the NOLH simulations when using the original blocking index of Schwierz et al. (2004a), because the PV anomalies are too weak, do not persist for more than 5 days, and/or are too mobile.

## 5  Conclusions

The relative roles of different processes for the formation and maintenance of atmospheric blocking have been debated for a long time (Woollings et al., 2018). While classical blocking theories are based on dry-adiabatic interactions of waves (e.g., Charney and DeVore, 1979; Shutts, 1983), the importance of moist-diabatic processes, in particular the release of latent heat in ascending airstreams, has recently been recognized to play a significant role in the dynamics of the upper-level large-scale flow,

including Rossby waves (e.g., Pomroy and Thorpe, 2000; Grams et al., 2011; Wirth et al., 2018) and blocking (Croci-Maspoli and Davies, 2009; Pfahl et al., 2015; Steinfeld and Pfahl, 2019; Müller and Névir, 2019). Motivated by this recent finding, the present study explores the effect of LH on the development of five different blocking cases with the help of sensitivity experiments with the ECMWF's global numerical weather prediction model IFS, in which cloud-related LH is altered in the storm track region upstream of the block.

A key finding of the numerical sensitivity experiments is that the intensity, spatial extent and lifetime of all simulated blocking events depends strongly on latent heating. In some cases (in 4 of 5 cases), the presence of LH even determines whether or not blocking (according to the blocking index of Schwierz et al. (2004a)) occurs at all. Consistent with the findings of previous studies (Davis et al., 1993; Stoelinga, 1996; Pauley and Smith, 1988; Pomroy and Thorpe, 2000), the primary effects of latent heating on the tropopause arise from the diabatic reduction of PV and the associated enhancement of the divergent outflow aloft. Latent heating accelerates the vertical motion and divergent outflow on the western flank of the block, locally by a factor of 4, and the succeeding interaction with the upper-level PV distribution modifies the amplification and propagation of upper-level waves and blocking compared to the simulations without latent heating. These processes act to slow down the eastward propagation and amplify the intensity and spatial extent of the negative PV anomaly in all cases.

A comparison between the five cases reveals a large case-to-case variability of the effect of latent heating on blocking, which depends strongly on the phase of the blocking life cycle and the state of the background flow. During the early growth phase, latent heating contributes to the initial ridge amplification and facilitates a faster growth of the incipient ridge. During the mature phase, on the other hand, the large-scale flow can further amplify also without the contribution of LH and thus appears to be less sensitive to changes in LH. This amplification is related to the state of the background flow: In the cases with a more meridional flow and a pre-existing large-scale ridge, a block also develops in the absence of latent heating, though weaker and less extended. The presence of this pre-existing ridge induces large-scale upper-level diffluent flow, which supports the meridional amplification of arriving synoptic-scale waves (eddy straining mechanism Shutts, 1983; Mullen, 1987) and the poleward quasi-adiabatic transport of low-PV air from lower latitudes ahead of baroclinic disturbances (e.g., Colucci, 1985). Nevertheless, as demonstrated in the case study of the maintenance of block Thor, the absence of latent heating can also lead to a more rapid decay of blocking. In this case, the dry-adiabatic forcing due to eddy straining in the diffluent region upstream of the block is not strong enough to sustain the system against dissipation.

While our experiments are limited to blocking situations, which are associated with a very strong large-scale flow amplification in the mid-latitudes, the diabatic formation of anticyclonic PV anomalies can be observed in various synoptic situations in which Rossby waves (e.g., Grams et al., 2011; Chagnon and Gray, 2015; Röthlisberger et al., 2018), cut-off lows and PV streamers (Knippertz and Martin, 2007; Madonna et al., 2014a) and Rossby wave breaking (Zhang and Wang, 2018) play a role. While in this study large changes, e.g., removal of LH, have been made to quantify the total effect of LH on blocking dynamics, previous studies demonstrated that also small changes to various parametrization schemes had an impact on a downstream ridge building (Joos and Forbes, 2016; Maddison et al., 2020). Additional sensitivity simulations with reduced and enhanced LH for the case Thor onset show that an increase in LH triggers nonlinear amplification in blocking with greater effect on spatial extent than on intensity. LH may therefore be dynamically relevant, influencing the jet stream and potentially the

downstream flow evolution in all these situations, which is likely to have important consequences for medium-range weather prediction.

The sensitivity experiments demonstrate that blocking is the result of a constructive interaction between diabatic heating and dry baroclinic processes. Intense latent heating occurs predominantly in the warm conveyor belt of extratropical cyclones (Wernli, 1997) and is thus in phase with and strongly coupled to the secondary circulation associated with dry adiabatic forcing (Kuo et al., 1990). Our sensitivity experiments corroborate earlier studies that the interaction between mobile synoptic-scale eddies and planetary-scale flow anomalies plays an important role for blocking formation and maintenance (Nakamura et al., 1997; Luo et al., 2014; Nakamura and Huang, 2018; Steinfeld and Pfahl, 2019), and show that diabatic processes can provide the required flow amplification in addition to dry-dynamical forcing. In order to properly represent blocking dynamics, numerical weather prediction and climate models thus have to correctly account for this coupling between dry and moist processes, including the details of microphysical processes that shape the spatial distribution of latent heating in clouds (e.g., Joos and Wernli, 2012; Dearden et al., 2016; Attinger et al., 2019).

*Code and data availability.* The blocking identification code CONTRACK is available from https://github.com/steidani/ConTrack. The code and information on how to use the Lagrangian Analysis tool LAGRANTO can be found from http://www.lagranto.ethz.ch. The data of the IFS sensitivity simulations is available from Daniel Steinfeld upon request.

*Author contributions.* S.P. and D.S. designed the study. D.S. performed the numerical experiments, analysed the data and wrote the paper. M.B., R.F. and S.P. provided guidance on interpreting the result. All authors commented on the manuscript.

*Competing interests.* The authors declare no conflict of interest.

*Acknowledgements.* D. Steinfeld acknowledges funding from ETH Research Grant ETH-09 15-2. M. Boettcher acknowledges funding from the Swiss National Science Foundation (grant no. 165941) and the European Research Council 485 (ERC) under the European Union's Horizon 2020 research and innovation programme (project INTEXseas, grant agreement No 787652). We thank Elisa Spreitzer, Roman Attinger, Hanna Joos, Michael Sprenger and Heini Wernli (ETH Zürich) and Christian Grams (KIT) for fruitful discussions on diabatic processes in mid-latitude weather systems. IFS model simulations have been performed as part of the ECMWF special project "Diabatic effects in mid-latitude weather systems". ECWMF and MeteoSwiss are acknowledged for making the IFS model available and for access to the ECMWF computing facilities. We also thank both reviewers for their constructive comments. The data analysis and visualization was done using Python.

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

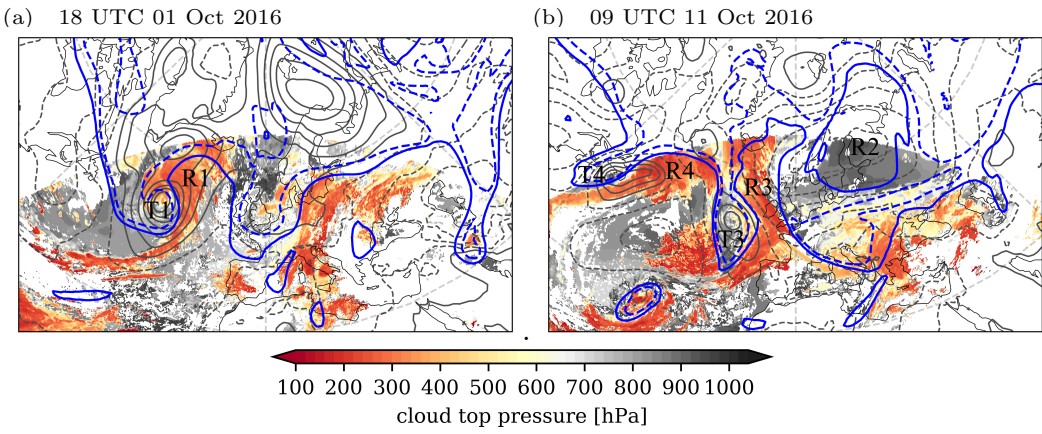

**Figure 1.** Synoptic situation over the North Atlantic at a) 18 UTC 01 Oct 2016 and b) 9 UTC 11 Oct 2016. SLP (gray contours, every 10 hPa, solid to dashed contours at 1015 hPa) and upper-level PV (blue contours, 2 (solid) and 3 (dashed) pvu) from the IFS CNTRL run after a) 42 hours lead time for Thor onset and b) 36 hours lead time for Thor maintenance. Labels "T1 - T4" mark troughs (cyclones) and "R1 - R4" mark ridges (anticyclones) and are described in the text. Cloud top heights (hPa, shading) from satellite imagery based on EUMETSAT MSG-SEVIRI data (EUMETSAT, 2017).

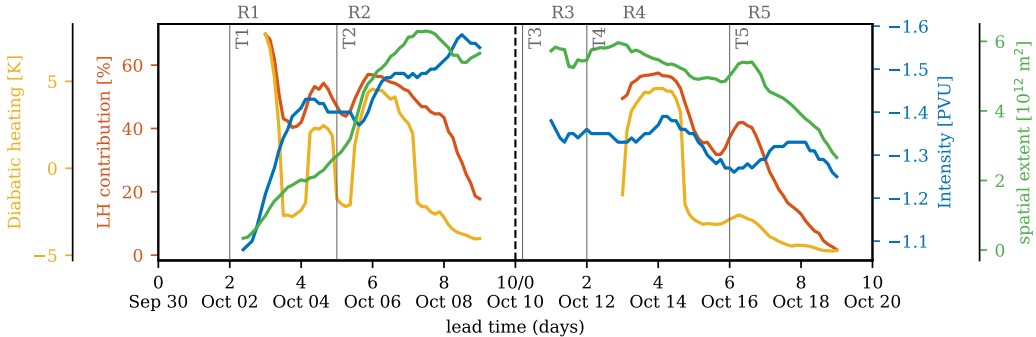

**Figure 2.** Percentage of trajectories with $\Delta\theta > 2\,\mathrm{K}$ in 3 days (red, %), mean diabatic heating along the blocking trajectories (yellow, K, calculated as the mean change in $\theta$ along all (heated and non-heated) trajectories), blocking intensity (blue, right axis, pvu), and spatial extent (green, 2nd right axis, $10^{12}\,\mathrm{m}^2$) as a function of time (simulation lead time and date) for Thor onset and maintenance. Note that 3-day backward trajectories can only be calculated after day 3. Labels "T1 - T5" and "R1 - R5" refer to the troughs and ridges during time of their interaction with block Thor. Note that no block is detected between 9–11 October as a result of the 2-day temporal smoothing of the upper-level PV anomaly field.

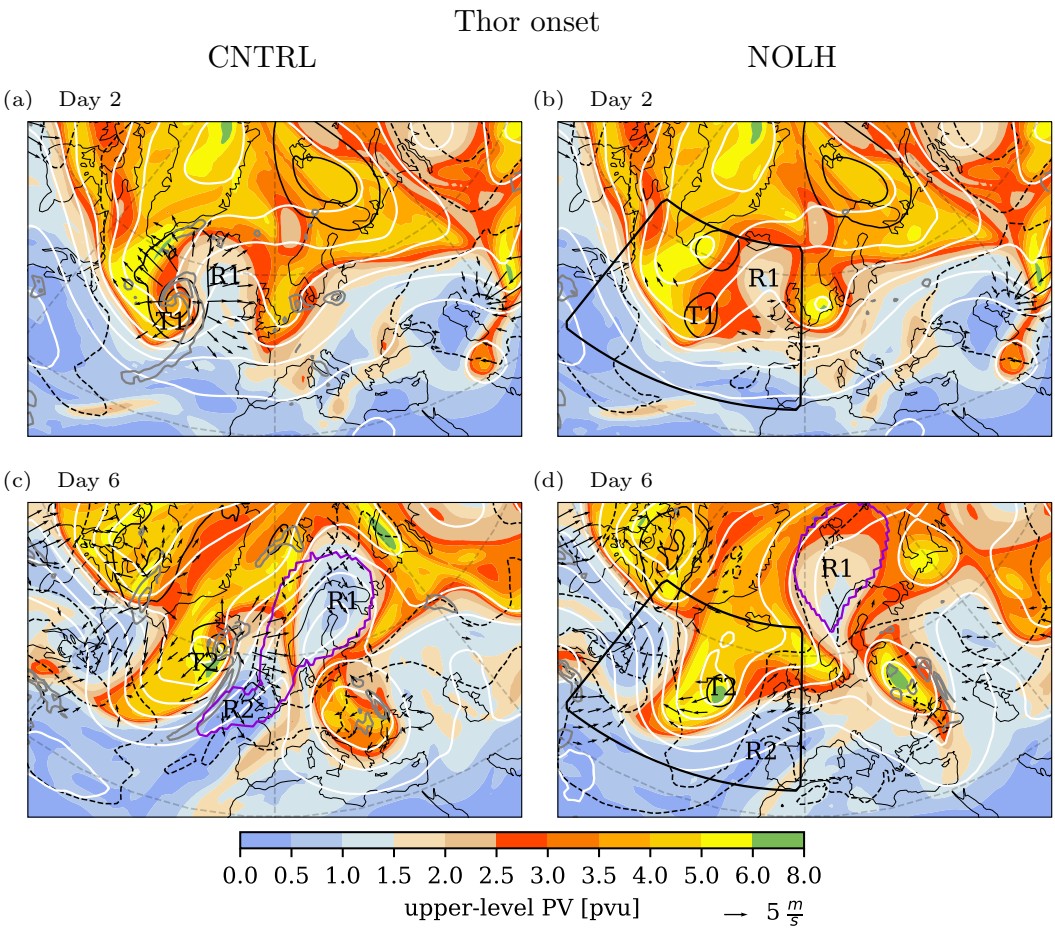

**Figure 3.** Upper-level PV (in pvu, shaded), upper-level divergent wind (black vectors according to reference vector, only shown for wind speed larger than $2\,\mathrm{m\,s^{-1}}$), geopotential height at $500\,\mathrm{hPa}$ (white contours every 100 gpm), cloud-diabatic (cloud microphysics and convection) heating (1 and $3\,\mathrm{K\,(3\,h)^{-1}}$ in gray contours, vertically integrated between 900 - 500 hPa), SLP (solid black contours from $1000\,\mathrm{hPa}$ every -10 hPa, dashed contours from $1020\,\mathrm{hPa}$ every +10 hPa), and blocking region (violet contour for PV anomaly of -1 pvu) in (left) CNTRL and (right) NOLH simulation at (a,b) 00 UTC 2 October 2016 (day 2) and (c,d) 15 UTC 5 October 2016 (day 6). Labels "T1 - T5" mark troughs (cyclones) and "R1 - R5" mark ridges (anticyclones) as described in the text. Black box in NOLH indicates region where LH is turned off.

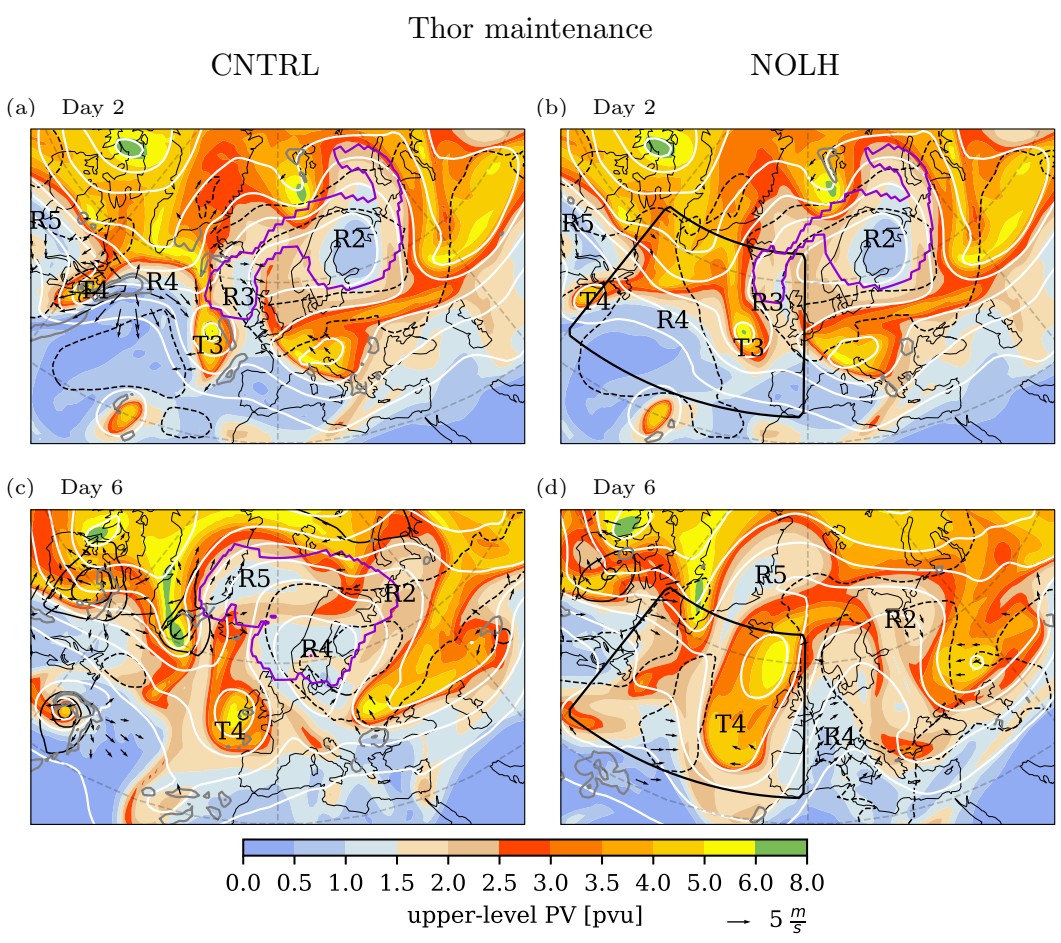

**Figure 4.** Same as Fig. 3, but at (a,b) 9 UTC 11 October 2016 (day 2) and (c,d) 9 UTC 16 October 2016 (day 6).

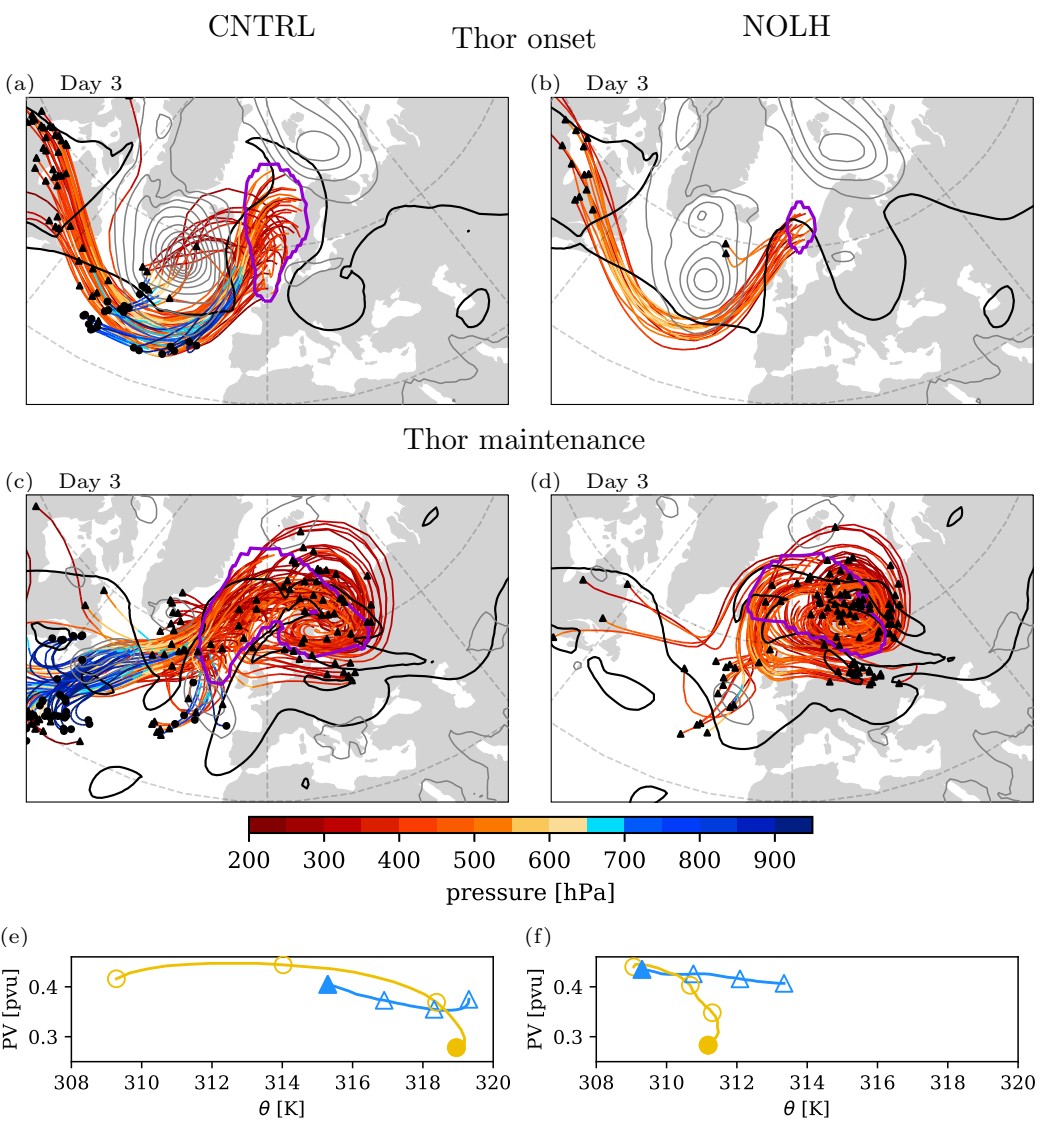

**Figure 5.** Upper-level 2 pvu contour (black line), SLP (gray contours, from 1000 hPa every -5 hPa) and blocking region (violet contour for upper-level PV anomaly of -1 pvu) for (a,c) CNTRL and (b,d) NOLH simulation for case Thor onset (upper panel) and Thor maintenance (middle panel). 72-h backward trajectories started in the blocking region at 00 UTC 4 October 2016 in the onset simulations and at 00 UTC 13 October 2016 in the maintenance simulations are shown as colored lines, with color indicating pressure (hPa). The black circles and triangles show the location of the heated ($\Delta\theta > 2\,\mathrm{K}$) and quasi-adiabatic ($\Delta\theta < 2\,\mathrm{K}$) trajectories 3 days prior to arrival in the blocking, respectively. (e,f) Median temporal evolution of $\theta$ and PV along heated (yellow) and quasi-adiabatic (blue) blocking trajectories for (e) CNTRL and (f) NOLH simulation. The evolution was calculated from all trajectories during the entire blocking lifetime of Thor onset and Thor maintenance cases. Filled markers show the median for each airstream at the time of the arrival in the blocking region, open markers show medians at days -1, -2, and -3 before arriving in the block.

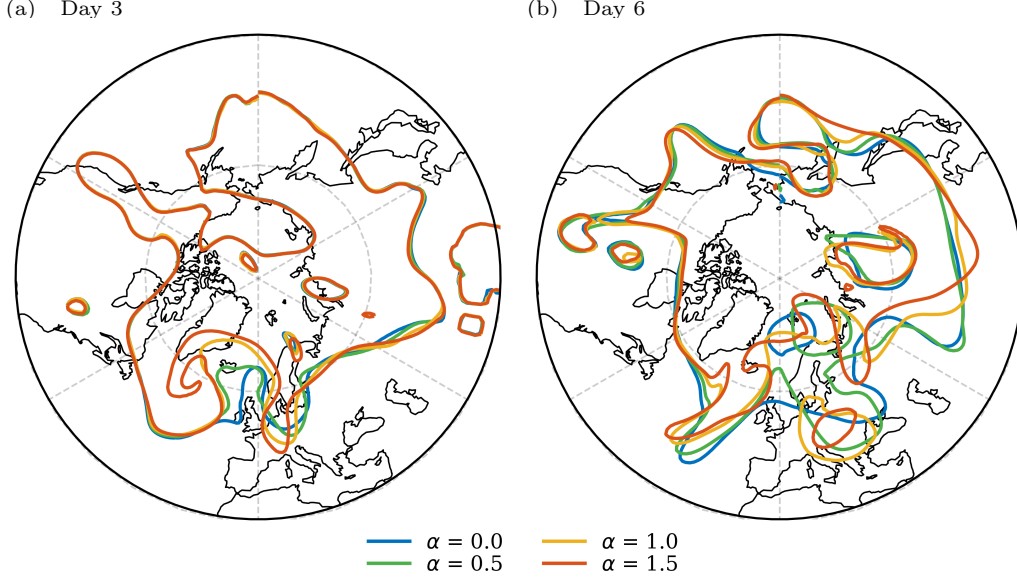

**Figure 6.** Dynamical tropopause (upper-level 2 pvu contour) for Thor onset during (a) 3 October 2016 (day 3) and (b) 6 October 2016 (day 6) for different $\alpha$ values (blue for $\alpha = 0$ (NOLH), green for $\alpha = 0.5$, yellow for $\alpha = 1$ (CNTRL), and red for $\alpha = 1.5$).

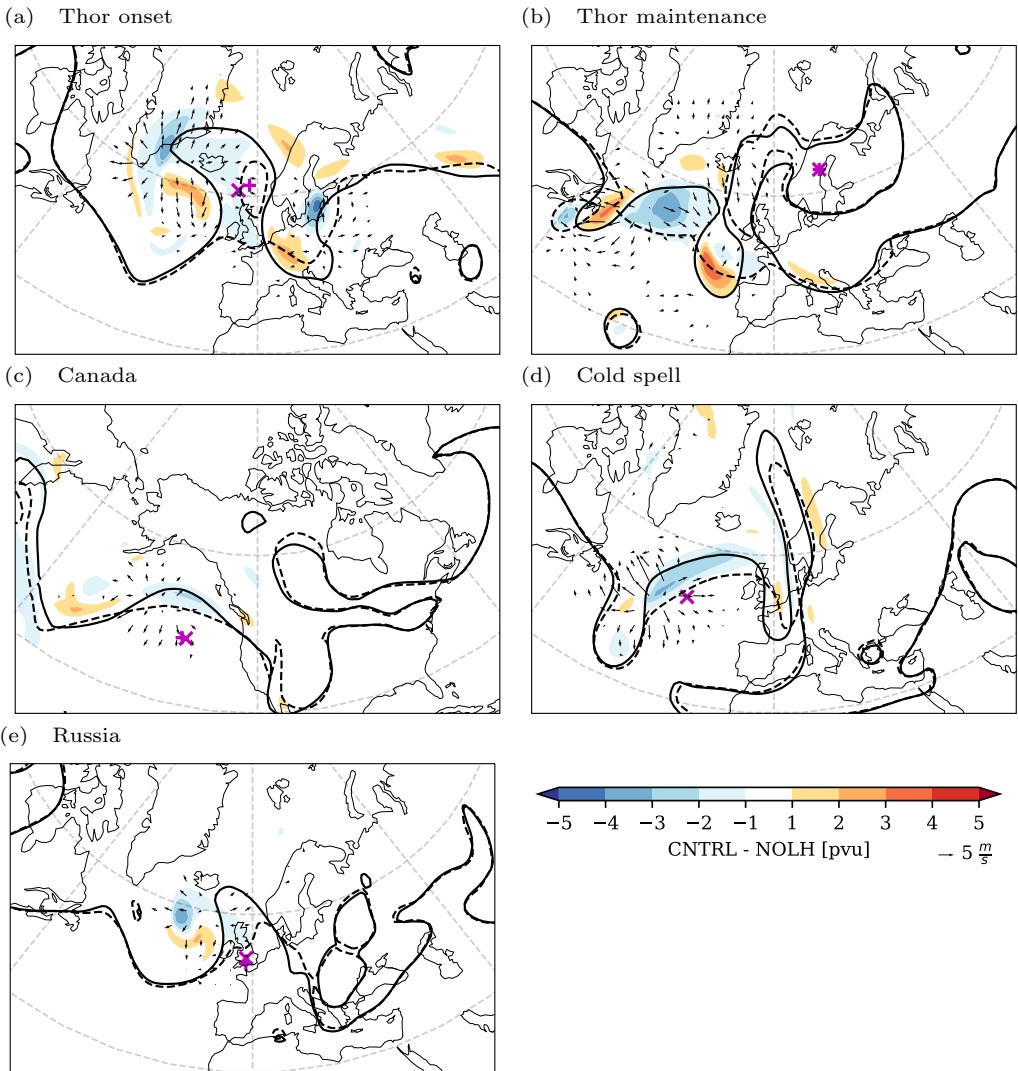

**Figure 7.** Difference (CNTRL - NOLH) in upper-level PV (shaded in pvu), difference in upper-level divergent wind (vectors only shown for wind speed larger than $1\,\mathrm{m\,s^{-1}}$), and upper-level 2 pvu contour (solid for CNTRL, dashed for NOLH) after 3 days model simulation for (a) Thor onset, (b) Thor maintenance, (c) Canada, (d) Cold Spell, and (e) Russia. "x" and "+" show locations of blocking centers for CNTRL and NOLH, respectively.

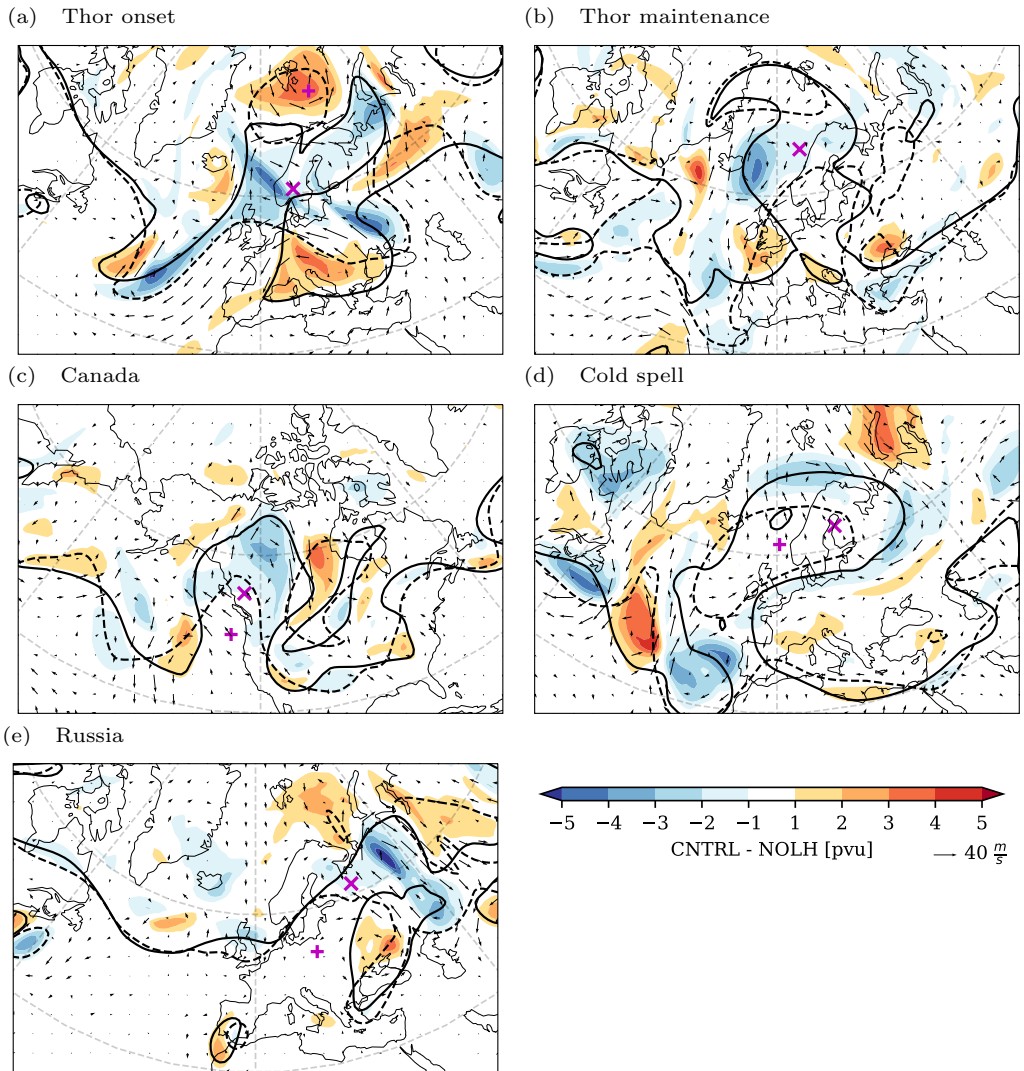

**Figure 8.** Difference (CNTRL - NOLH) in upper-level PV (shaded in pvu), difference in upper-level rotational wind (vectors only shown for differences larger than $1\,\mathrm{m\,s^{-1}}$), and upper-level $2\,\mathrm{pvu}$ contour (solid for CNTRL, dashed for NOLH) after 6 days model simulation for (a) Thor onset, (b) Thor maintenance, (c) Canada, (d) Cold spell, and (e) Russia. "x" and "+" show locations of blocking centers for CNTRL and NOLH, respectively.

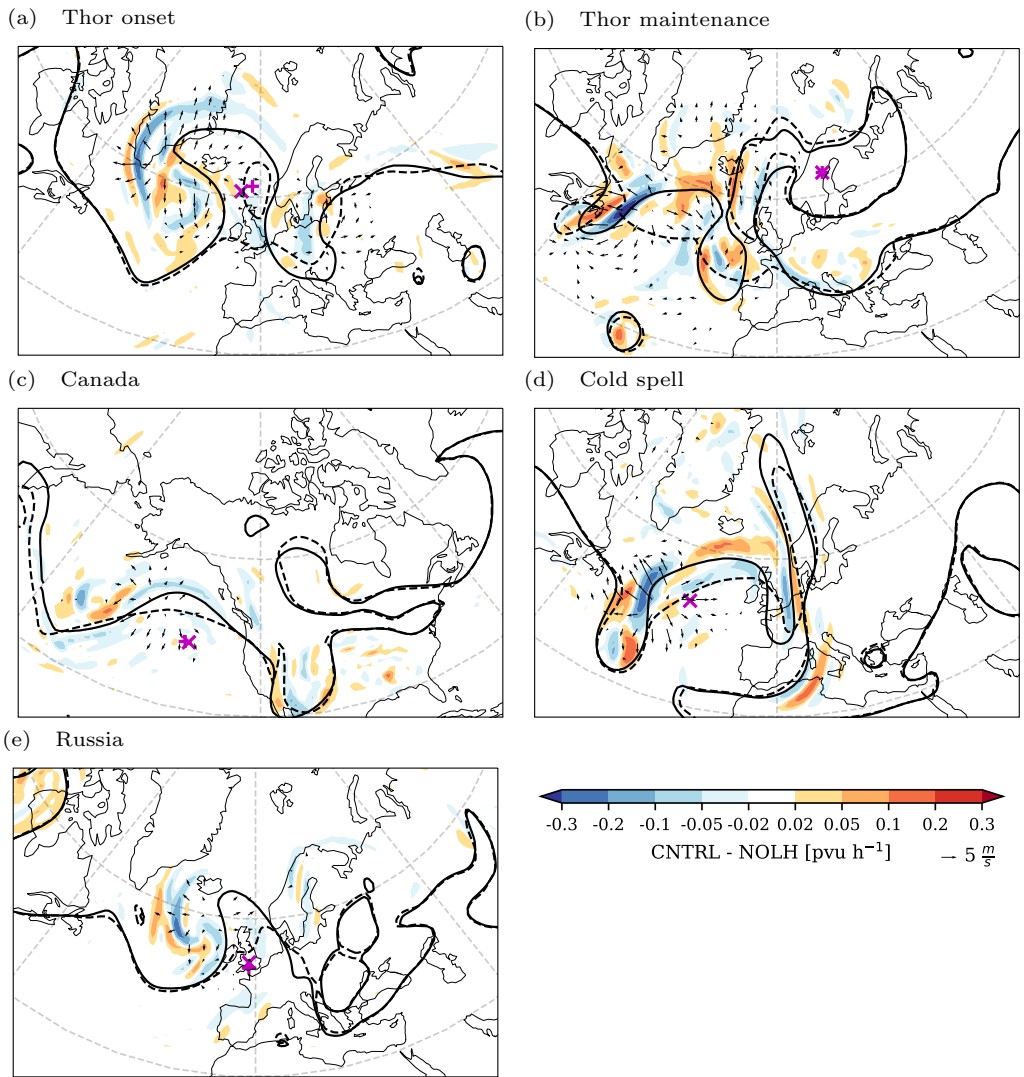

**Figure 9.** Difference (CNTRL - NOLH) in upper-level PV advection by the divergent (irrotational) wind ($\mathbf{v}_\chi \cdot \nabla PV$, shaded in $\mathrm{pvu\,h^{-1}}$), difference in upper-level divergent wind (vectors only shown for wind speed larger than $1\,\mathrm{m\,s^{-1}}$), and upper-level 2 pvu contour (solid for CNTRL, dashed for NOLH) after 3 days model simulation for (a) Thor onset, (b) Thor maintenance, (c) Canada, (d) Cold Spell, and (e) Russia. "x" and "+" show locations of blocking centers for CNTRL and NOLH, respectively.

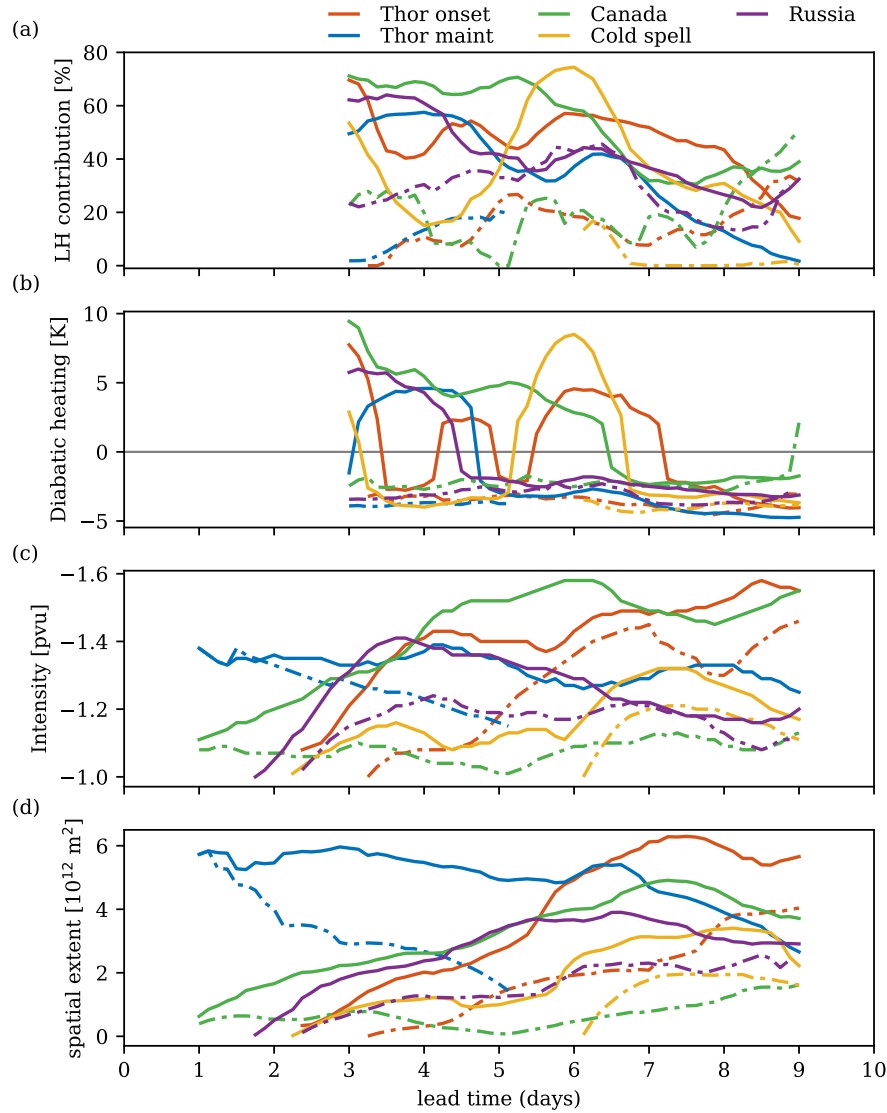

**Figure 10.** (a) Percentage of blocking trajectories with $\Delta\theta > 2$ K in 3 days (%), (b) mean diabatic heating (K, calculated as the mean change in $\theta$ along all (heated and non-heated) trajectories), (c) blocking intensity (upper-level PV anomaly), (d) spatial extent ($10^{12}$ m$^2$) as a function of simulation lead time. Solid lines for CNTRL simulations, dashed lines for NOLH simulations. Note that the individual curves start as soon as a block is identified with the PV-anomaly index, and 3-day backward trajectories can only be calculated after day 3.

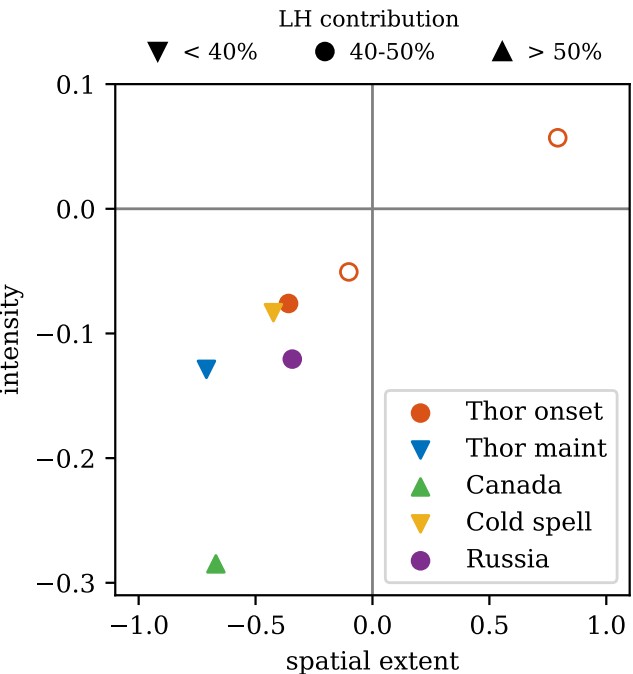

**Figure 11.** Normalized difference in peak spatial extent and peak intensity of the NOLH blocks compared to the CNTRL blocks. Values close to zero indicate weak sensitivity. Markers indicate the LH contribution in the CNTRL simulations (see Table 1). Red open circles for Thor onset simulations with reduced LH ($\alpha = 0.5$) and enhanced LH ($\alpha = 1.5$).

**Table 1.** Selected historical blocking events. The LH contribution has been determined from backward trajectory calculations. The initialization time is the same for both CNTRL and NOLH simulations. Note that "Thor onset" and "Thor maintenance" are different phases of the same blocking event.

| Experiment | Flow pattern | Initiation time | Region | NOLH box | LH contribution | |
|---|---|---|---|---|---|---|
| | | | | | CNTRL | NOLH |
| Russia | omega | 29 June 2010 | Western Russia | [60°W - 0°, 35°N - 65°N] | 42 % | 29 % |
| Canada | omega | 27 Apr 2016 | Pacific-America | [180°W - 120°W, 35°N - 65°N] | 52 % | 20 % |
| Thor onset | dipole | 30 Sep 2016 | Atlantic-Europe | [60°W - 0°, 35°N - 65°N] | 47 % | 16 % |
| Thor maintenance | dipole | 10 Oct 2016 | Atlantic-Europe | [60°W - 0°, 35°N - 65°N] | 34 % | 12 % |
| Cold spell | dipole | 18 Feb 2018 | Atlantic-Europe | [60°W - 0°, 35°N - 65°N] | 38 % | 3 % |