# Peer review of "The sensitivity of atmospheric blocking to upstream latent heating numerical experiments"

_Weather and Climate Dynamics, 2020_

## Referee Comment (RC1) · Oscar Martinez-Alvarado (Referee) · 20 Mar 2020

General comments

Building upon recent findings related to the importance of tropospheric latent heating on the development of atmospheric blocking, this contribution investigates from a numerical modelling point of view, the extent to which latent heating influences the development of atmospheric blocking and the cause-and-effect relationship involved in this influence. Understanding these processes in the atmosphere is critical due to the important effects that blocking has at the surface and on human activities. Thus, these are without doubt relevant scientific questions within the scope of WCD. For this

investigation the researchers performed sensitivity analysis by varying latent heating in ad-hoc regions in numerical simulation of five cases, using the state-of-the-art ECMWF IFS and an advanced methodology based on atmospheric blocking tracking and trajectory analysis. Through their investigation they demonstrate in a convincing manner that atmospheric blocking features such as intensity, spatial extent and lifetime depend strongly on latent heating. However, they also showed that there is a large case-to-case variability. The paper is very well structured and written, and, in my opinion, the description of the methodology is sufficiently complete to allow their reproduction by fellow scientists. Therefore I recommend the article for publication in Weather and Climate Dynamics. I include a list of minor comments that could be considered by the authors to hopefully enhance the paper.

Specific comments

L66: How smooth are the physical temperature tendencies in the native resolution? If it is not a smooth field, is it properly represented after the interpolation to the 1-degree horizontal resolution?

L80-81: Please cite the previous studies that the methodology in this study is being contrasted against? In which way is the new methodology different to the one in previous studies? Did they dampened latent heating everywhere in their domain?

L105-106: How was the blocking event for which latent heating was reduced and increased chosen? Is the case representative in any way especially after considering that large case-to-case variability reported in this study?

L279-280: Is there any indication of the extent of the influence of initial conditions on the differences after 6 days? How would the differences found here compare to differences between members in an ensemble simulation? This is discussed to a certain extent in Maddison et al. (2020, doi:10.1002/qj.3739), which is in any case a relevant reference that you might want to cite.

L385-389: The Russia block is very interesting, and the discussion could be extended. If there is such a limited influence of latent heating in the evolution of the block, what is then the source of the big differences in the evolution of the blocks in the two simulations?

L411: I've got a bit confused with this description, in which the authors talk about a median heating of 3 K (dashed curves in Fig. 10a,b). What I can see is cooling in those curves? I'm sure I'm missing something. Can you clarify?

L435-436: Should the statement that the Thor onset and Cold spell block amplify without the contribution of LH be qualified? The LH was eliminated only between 900 - 500 hPa, and as the authors acknowledge in Section 2 there are diabatic processes active above that layer.

L464: Should the intensity of the upstream cyclone be included in the list of factors as is done in L432-433?

Technical corrections

L58-59: Delete 'exemplarily' or change it for 'as an example' after 'introduces'

L65: In addition to the number of vertical levels, give details on the top of the atmosphere and the typical separation between levels.

L115: Change 'quasi-stationary' for 'quasi-stationarity'

L123: Spatial extent is among the set of blocking characteristics calculated for each blocking event. Even though the method to identify blocking considers an atmosphere's layer rather than a single level, the extent referred to here is horizontal extent rather than a three-dimensional size. Is this so? It would be useful to add details on the layer considered for the blocking identification. Is it the 'upper-level' layer, i.e. 500 - 150 hPa? Is it possible to compute details on the vertical extent of the blocking region?

L230 and 241: There are references to Fig. 3e,g, but I cannot see those panels.

L220-221: Where are the trajectories emanating from in the vertical direction? Are they initially located between 500 hPa and 150 hPa? Or at a particular level?

L269: Are the divergent wind speeds quoted averages over a region? Please, specify.

L392: Add 'simulations' after '... NOLH (dashed lines)'.

---

## Referee Comment (RC2) · Florian Pantillon (Referee) · 24 Mar 2020

Review of "The sensitivity of atmospheric blocking to changes in upstream latent heating – numerical experiments" by Daniel Steinfeld, Maxi Boettcher, Richard Forbes, and Stephan Pfahl.

The paper investigates the contribution of latent heating during the onset phase of blocking in four case studies spanning the North Hemisphere. The investigation is extended to the maintenance phase for one case study, which is described more thoroughly. The contribution of latent heating is quantified by switching off heating related to cloud processes in a region located upstream of the blocking in sensitivity experi-

ments with the global IFS model. The impact is diagnosed using the potential vorticity anomaly and divergent wind at upper levels mainly. The results show a clear contribution of latent heating, including periods of bursts, to the intensity of blockings and their extent in space and time, with large case-to-case variability that appears to depend on the flow configuration.

The paper addresses an important topic in atmospheric dynamics, is based on well-designed numerical experiments, and is well written overall. However, as detailed in the general comments below, it contains majors flaws related to a lack of assessment of the numerical experiments, a lack of balance between one detailed case study and three quicker ones, and a general lack of consistency between text and figures. Although the paper is definitely interesting and valuable, it gives a feeling of subjectivity in the choice of case studies and interpretation.

Considering that a systematic analysis of all presented case studies would require much additional work, and as the discussion at the end of Section 4 currently suggests that the impact of latent heating depends on many parameters that cannot be properly covered here, I suggest to remove the additional cases altogether and focus on the Thor case more thoroughly. For instance, with less extra work, the additional sensitivity experiments with alpha=0.5 and 1.5 could be included in Figs. 9–10 to discuss non-linearity, or the respective contribution of microphysics and convection to latent heating could be quantified to contribute to the current discussion in the NAWDEX community.

General and specific comments are listed below to help improving the paper.

GENERAL COMMENTS

I. An assessment of the quality of control simulations is lacking: Fig. 1 provides some comparison with satellite observations for the Thor case but in an indirect fashion and at short range only, while nothing is provided for the other cases. A 10-day run is not expected to perfectly match observations but needs to capture the blocking at least. The lack of predictabillity during the onset of blocking makes this questionable. An

easy solution would be to add panels for the analysis fields on Figs. 3–4 (and S1–S3) and curves on Figs. 2 and 6.

II. The organization is unbalanced: most of the paper is dedicated to the case study of Thor but related contents are spread between Sections 3 (which contains a subsection 3.1 without 3.2) and 4.1 with some repetitions in 4.2, while additional cases are briefly introduced in 2.4 (without motivation) then discussed in Section 4.2 only (without prior description of their specific dynamics). Describing one case study in details and several cases succinctly is a sound approach but in the present form I do not clearly see what to learn from these additional cases.

III. All along the paper, features such as the upper-level jet stream are discussed but not shown anywhere, while striking contrasts between case studies are not mentioned. Please make sure you actually display what you describe and describe what you display. And please avoid wording such as "it is evident", in particular for statements that are not.

SPECIFIC COMMENTS

The title could be more specific: "changes" is vague

l. 3 "the causal relationship between latent heating and blocking formation has not yet been fully elucidated": what is the paper's contribution to elucidating this causal relationship? (which likely extends beyond blocking "formation" only)

l. 8–12 This does not reflect the contents of the paper: "the jet stream" is not shown anywhere; "warm conveyor belt airstreams" are barely discussed; "an accurate parameterization of microphysical processes" is not particularly supported by the results.

l. 31–33 please develop "the mechanism behind the classical view"

l. 47–49 why are diagnostic methods not able to show a causal relationship?

l. 64–65 This is lower resolution than the operational version and previous studies

suggested that LH is sensitive to resolution: can you compare your control simulations with the operational IFS forecast to estimate this sensitivity?

l. 78 LH is turned off for both microphysics and convection schemes but convective motions are not captured by backtrajectories: what is their contribution?

l. 140 typo?

l. 154–160 It is unclear why these specific cases were selected

l. 165–167 For clarity, and because the nomenclature is used by Maddison et al. (2019),"Stalactite Cyclone" should be mentioned here

l. 171–179 and Fig. 1 The paragraph needs improvement: (1) it is not "evident" to recognize the mentioned features, esp. at mesoscale (please zoom in and/or mark them); (2) a visual comparison between "upper-level" PV (defined as 500–150 hPa mean, please remind in the caption) and cloud top pressure (which is not directly "observed" by MSG) is not "quantitative"; (3) comparisons are for short lead times (36h and 42h, which should be indicated) thus do not support that the evolution is "well predicted" in the 10-day simulations and contradict the "large forecast uncertainty" mentioned above.

l. 186 what is the APV "index" exactly?

l. 190 "confirm": is it expected?

l. 191 why are quasi-adiabatic processes associated with cooling?

l. 196–197 this is slightly below average compared to the climatology cited above

l. 207–208 is this shown somewhere? It is not obvious. . .

l. 210–219 The discussion is hard to follow, as the ingredients are not explicitly shown ("jet splitting", "deformation region", "poleward transport", "ex-tropical cyclone", "migratory ridge"). Either detail and add information on Fig. 4, or streamline.

l. 226 how many is "many"?

l. 230 "Fig. 3 a, c, e, g": do you mean Figs. 3 a, c and 4 a, c? Only 3a and 4a are related to trajectories shown in Fig. 5a, c.

l. 241 see comment above

l. 245, 247 "quasi-adiabatic": did you explicitly check the 2K heating criterion or do you refer to the stable pressure along trajectories?

l. 256 "mid-level": better lower-level in contrast with upper-level for 500–150 hPa?

l. 258 "initial time steps" is confusing for day 2: better early evolution?

l. 270 "cold front" and l. 272, 285 "jet stream": are these features shown somewhere?

l. 285 "as a consequence": I am not sure this is due to R2 only as the wave pattern is modified altogether (see l. 280)

l. 291, 295 "deformation flow", "diffluent flow": not shown?

l. 296–297 see l. 285

l. 298 is cooling explicitly computed along trajectories?

l. 301 Fig. 4 c, d

l. 311–312 see l. 270, 291, 295, . . .

l. 322–323 again, what is the motivation for selecting these specific cases?

l. 331–332 not really: (1) there is substantial case-to-case variability and (2) differences cannot be attributed to "the upstream cyclone" if it is neither showed nor mentioned for the additional cases

l. 339–342 is this all shown somewhere or suggested only? Fig. S4 does not include the additional cases.

l. 350 where is the dipole pattern? Please indicate (a), (b), (c), etc.

l. 355–361 this description (and the related panels) must be moved to the Thor Section above.

l. 369–376 That is certainly interesting but I do not know where to see all of it in Fig. 8.

l. 380–382 Does the box move with the strongest ascent/divergence/advection? Can you show an example? On Fig. 8 in particular it is not obvious where it would be placed.

l. 383–385 again, there is substantial variability both between cases and between lead times (weak signal beyond one week for instance)

l. 385 "magenta": rather violet?

l. 386–387 is this shown somewhere?

l. 396 sorry to insist but there is again case-to-case variability: the Canada case does not show bursts

l. 396–409 the discussion is not supported by any material: cyclones are not shown in the figures but for the case of Thor

l. 401 which trajectories exactly? Cooling is not explicitly shown in Fig. 5.

Fig. 2 spatial "extent"; labels T1–T5 appear to refer to troughs rather than cyclones.

Figs. 7 why focus on day 3 here and not on day 2 as above?

---

## Author Comment (AC1) · 28 Apr 2020

We thank all reviewers for their many critical and constructive comments. Please see the attached supplement for our detailed responses.

Please also note the supplement to this comment:
http://www.weather-clim-dynam-discuss.net/wcd-2020-5/wcd-2020-5-AC1-supplement.pdf

---

## Author Response (AR1)

**Author response to referee comments in manuscript wcd-2020-5:**

**"The sensitivity of atmospheric blocking to changes in upstream latent heating – numerical experiments"**
**by Daniel Steinfeld, Maxi Boettcher, Richard Forbes, and Stephan Pfahl**

We would like to thank both reviewers, Oscar Martinez-Alvarado and Florian Pantillon, for their positive, detailed and constructive feedback that helped us to improve the quality of our manuscript. The major changes in the new version of the manuscript are the following:

1. The introduction am method section has been restructured: the motivation, scope and objectives are explained more clearly.
2. A description of the synoptic evolution of the other three cases (in addition to Thor) is included.
3. The figures and corresponding references in the text have been improved.
4. The description of the case-to-case variability has been improved, including a new figure.

Our point-by-point responses follow below, in blue, with the original referee comments shown in black. A marked-up manuscript version showing the changes made can be found after the responses.

**1 Response to Florian Pantillon**

**Referee comment**

The paper investigates the contribution of latent heating during the onset phase of blocking in four case studies spanning the North Hemisphere. The investigation is extended to the maintenance phase for one case study, which is described more thoroughly. The contribution of latent heating is quantified by switching off heating related to cloud processes in a region located upstream of the blocking in sensitivity experiments with the global IFS model. The impact is diagnosed using the potential vorticity anomaly and divergent wind at upper levels mainly. The results show a clear contribution of latent heating, including periods of bursts, to the intensity of blockings and their extent in space and time, with large case-to-case variability that appears to depend on the flow configuration. The paper addresses an important topic in atmospheric dynamics, is based on well-designed numerical experiments, and is well written overall. However, as detailed in the general comments below, it contains majors flaws related to a lack of assessment of the numerical experiments, a lack of balance between one detailed case study and three quicker ones, and a general lack of consistency between text and figures. Although the paper is definitely interesting and valuable, it gives a feeling of subjectivity in the choice of case studies and interpretation. Considering that a systematic analysis of all presented case studies would require much additional work, and as the discussion at the end of Section 4 currently suggests that the impact of latent heating depends on many parameters that cannot be properly covered here, I suggest to remove the additional cases altogether and focus on the Thor case more thoroughly. For instance, with less extra work, the additional sensitivity experiments with alpha=0.5 and 1.5 could be included in Figs. 9–10 to discuss non-linearity, or the respective contribution of microphysics and convection to latent heating could be quantified to contribute to the current discussion in the NAWDEX community.

General and specific comments are listed below to help improving the paper.

**General comments**

1. An assessment of the quality of control simulations is lacking: Fig. 1 provides some comparison with satellite observations for the Thor case but in an indirect fashion and at short range only, while nothing is provided for the other cases. A 10-day run is not expected to perfectly match observations but needs to capture the blocking at least. The lack of predictability during the onset of blocking makes this questionable. An easy solution would be to add panels for the analysis fields on Figs. 3–4 (and S1–S3) and curves on Figs. 2 and 6.

   **Reply** Thanks for the discussion about the quality of the control simulations. We agree that an assessment of the performance of NWP models during blocking situations is an important aspect, especially since blocking is notoriously difficult to forecast (e.g. Tibaldi and Molteni, 1990; Pelly and Hoskins, 2003; Jung, 2014; Matsueda, 2009, Quandt et al., 2017). There are recent studies focusing on the role of WCBs in NWP forecasts during blocking onset (Grams et al., 2018, Maddison et al., 2019 and 2020). However, the predictability and forecast performance (of the control simulations) is not in the scope of this study, and a more detailed discussion would dilute its focus. Our conclusions on the large-scale impact of LH on blocking dynamics (comparison of CNTRL versus NOLH simulations) do not depend on the realistic reproduction of the selected blocking cases in 10-day forecasts, as we compare simulations to simulations. The only requirement is that the reference simulations capture large-scale blocking conditions, which we have demonstrated and quantified in the manuscript. Note that the role of LH for atmospheric blocking has only been recently discussed (Pfahl et al., 2015), and its contribution is still debated in the blocking community (Woolings et al., 2018; Voosen, 2020). Our results contribute to the current debate.

   In favour of keeping the paper focused, we decided not to include an assessment of the forecast quality. We improve the explanation of the motivation, open questions and scope of this study in the introduction.

   As the other reviewer also asked about the forecast quality of the control simulations, we still comment on this aspect here.

   All control simulations are initialized during the intensification phase of the upstream cyclone, which is typically 2 days prior to blocking onset. The IFS initialization is based on two requirements: (1) LH has to be removed early enough to ensure that its contribution to the ridge amplification is minimal and (2) the control simulation needs to capture the development of a major block, thus the initiation time has to be close to the onset. As an example, for "Thor onset" this is the 30 Sep 2016. This is in contrast to Maddison et al. (2019, 2020), who initiated the ensemble forecast on 27 and 28 Sep 2016, which leads to considerable divergence of the ensemble members at the time of blocking onset.

   Figure AR1 compares upper-level PV between the control simulation (CNTRL) and the operational analysis fields (ANA) for Thor onset. On 3 October 2016, 3 days into the forecast evolution (Fig. AR1a) the ridge amplification (onset of Thor) is very well represented. By 6 October, 6 days into the forecast, the anticyclonic wave breaking and the intensity and spatial extent of Thor is generally well represented in CNTRL (AR1b). However, there is an eastward shift of the block in CNTRL compared to ANA. Nevertheless, the forecast evolution of the block in CNTRL is similar enough to reality over the time of interest and captures an intense dipole block over Europe, and therefore allows studying the impact of LH on the flow amplification in the IFS sensitivity experiments.

[Figure]

a) after 3 days

Date: 20161003_00

b) after 6 days

Date: 20161006_00

CNTRL - ANA [pvu h$^{-1}$]

Figure **AR1**: Difference (CNTRL – ANA) in upper-level PV (shaded in pvu) and upper-level 2 pvu contour (solid for CNTRL, dashed for ANA) after a) 3 and b) 6 days model simulation for Thor onset.

2. The organization is unbalanced: most of the paper is dedicated to the case study of Thor but related contents are spread between Sections 3 (which contains a subsection 3.1 without 3.2) and 4.1 with some repetitions in 4.2, while additional cases are briefly introduced in 2.4 (without motivation) then discussed in Section 4.2 only (without prior description of their specific dynamics). Describing one case study in details and several cases succinctly is a sound approach but in the present form I do not clearly see what to learn from these additional cases.

   **Reply** We realise that we failed to explain the motivation for the selection of the different cases. The other blocking cases were selected because they were associated with extreme weather events that had a strong socio-economic impact. In addition, the 5 cases cover the typical range of different flow configuration (omega versus dipole block) during different seasons and with different LH contribution:
   - 2010 Russia (omega block, summer): Heat wave and forest fire in Russia
   - 2016 Canada (omega block, spring): Heat wave and devastating wildfire in Alberta
   - 2018 Cold Spell (dipole block, winter): Cold spell over Europe, with heavy snow in England ("beast from the east")
   - 2016 Thor (dipole block, autumn): during NAWDEX, onset and maintenance phase

Of course, such a selection of a limited number of case studies is somewhat subjective. Nevertheless, we think that the (briefer) analysis of additional blocking cases is a key part of our study as it demonstrates the strong case-to-case variability of the impact of latent heating on blocking, a point that the referee also mentions in his introductory statement. As such, the paper goes beyond a single case study, but having the same level of detail as for Thor would increase the length of the paper too much.

We believe that a detailed analysis of Thor in the beginning is meaningful as an illustrative example of blocking, its interaction with different weather systems, and as a transition to the sensitivity experiments in section 4.2 ("sets of blocks"). For clarification, we added a statement at the beginning of section 4 that case Thor will be further discussed in the next section together with the other cases. We try to improve the structure and avoid repetition between Section 4.1 and 4.2 as much as possible.

We included a paragraph in which we introduce the other cases with a short synoptic description. We improved the discussion about the case-to-case variability, as it is a key result, and tried to be more explicit about the differences (LH contribution,

sensitivity related to size and intensity) and similarities (intense cyclogenesis during onset, development of a cut-off anticyclone in NOLH) between the cases. For that, we will provide a new Figure 11, in which each case is shown as a dot (CNTRL and NOLH during mature phase) in a phase space of intensity versus spatial extent (similar to Figure 8 in Maddison et al., 2020).

[Figure]

New Figure **11**: Normalized difference in peak spatial extent and peak intensity of the NOLH blocks compared to the CNTRL blocks. Values close to zero indicate weak sensitivity. The size of the marker indicates the LH contribution in the CNTRL simulations (see Table 1). Red open circles for Thor onset simulations with reduced LH (α= 0.5) and enhanced LH (α= 1.5).

3. All along the paper, features such as the upper-level jet stream are discussed but not shown anywhere, while striking contrasts between case studies are not mentioned. Please make sure you actually display what you describe and describe what you display. And please avoid wording such as "it is evident", in particular for statements that are not.
   **Reply** Thanks for your suggestion. We adjusted the figures (added SLP to show surface cyclones), revised the text and more explicitly refer to the figures in the revised manuscript (including labels and locations of blocking center), which helps identifying the discussed features. However, from our general understanding it is sufficient to show upper-level PV (dynamical tropopause) and Z500 gradients to qualitatively describe the large-scale flow / jet steam, as the region of strongest wind follows the band of enhanced PV/Z500 gradient.

**Specific comments**

4. The title could be more specific: "changes" is vague
   **Reply** One alternative is «The sensitivity of atmospheric blocking to upstream latent heating - numerical experiments". It emphasize the first-case importance of upstream LH in blocking dynamics and highlights the novelty of this sensitivity study.

5. l. 3 "the causal relationship between latent heating and blocking formation has not yet been fully elucidated": what is the paper's contribution to elucidating this causal relationship? (which likely extends beyond blocking "formation" only)
   **Reply** Previous studies based on (Lagrangian) diagnostics have established a correlation between blocking and upstream LH. However, they are not able to directly show whether LH has a causal effect and critically modifies the development of blocking (Would a block still develop without LH?). Here, by removing upstream LH in sensitivity experiments, we demonstrate, for the first time, the cause-and-effect relationship between LH and blocking. The experiments contribute to the current debate about the role of LH in blocking. The results may be extended to other

situation of strong ridge amplification, as climatological studies have shown strong WCB activity prior to Rossby wave initiation (Röthlisberger et al., 2018) and wave breaking (Zhang and Wang, 2018). We added the following regarding these potentially broader implications to the manuscript: "While our experiments are limited to blocking situations, which are associated with a very strong large-scale flow amplification in the mid-latitudes, the diabatic formation of anticyclonic PV anomalies can be observed in various synoptic situations in which Rossby waves (e.g., Grams et al., 2011; Chagnon and Gray, 2015; Röthlisberger et al., 2018), PV streamers, cut-off lows (Knippertz and Martin, 2007; Madonna et al., 2014) or wave breaking (Zhang and Wang, 2018) play a role. LH may therefore be dynamically relevant, influencing the jet stream and potentially the downstream flow evolution in all these situations, which is likely to have important consequences for medium-range weather prediction."

6. l. 8–12 This does not reflect the contents of the paper: "the jet stream" is not shown anywhere; "warm conveyor belt airstreams" are barely discussed; "an accurate parameterization of microphysical processes" is not particularly supported by the results.
**Reply** Thanks for the comment. We quantify and discuss more carefully WCB trajectories. See also reply to general comment 3. Instead of «microphysical processes», we put «moist processes in ascending airstreams» which is a bit wider and covers convection, clouds and WCBs.

7. l. 31–33 please develop "the mechanism behind the classical view"
**Reply** The classical view describes the interaction between transient eddies and their positive feedback on blocking maintenance. More specifically, it describes the thermal and vorticity advection ahead of synoptic waves that experience straining and slow down in a diffluent flow (Shutts, 1983; Yamazaki and Itho, 2013). From a Lagrangian viewpoint, this is linked to the quasi-adiabatic advection of low-PV air polewards into the blocking region, which is captured by our trajectory-based diagnostic ($\Delta\theta < 2K$) (see also Steinfeld and Pfahl, 2019).
In the revised manuscript, we put our results into perspective with these more traditional blocking concepts.

8. l. 47–49 why are diagnostic methods not able to show a causal relationship?
**Reply** See our reply to comment 5. Previous trajectory analysis indicated that LH is often present upstream of blocking, but this doesn't necessarily imply that it has a strong causal impact on blocking dynamics.

9. l. 64–65 This is lower resolution than the operational version and previous studies suggested that LH is sensitive to resolution: can you compare your control simulations with the operational IFS forecast to estimate this sensitivity?
**Reply** We think that for the purpose of this study the chosen resolution is adequate, as the simulations capture the development of a large-scale blocking flow, and our main conclusions are based on the comparison of different model runs with the same resolution. In principle, the effects of LH could be different when smaller-scale convective motion was resolved, but running all our experiments with convection-permitting resolution is not feasible (recall that also a large model domain is required). See our reply to general comment 1 why we do not include an assessment of the forecast quality in the revised manuscript.

10. l. 78 LH is turned off for both microphysics and convection schemes but convective motions are not captured by back trajectories: what is their contribution?
**Reply** As also mentioned in our previous reply, it is a limitation of our study that trajectories follow the resolved large-scale wind and do not directly capture

convective ascent. This might lead to an underestimation of the relevance of LH. We discuss the limitations more carefully.

Latent heat release due to the convection scheme still contributes to the cross-isentropic ascent of the trajectories. Figure AR2 shows the contribution of cloud microphysics and convection integrated along each heated trajectory ($\Delta\theta > 2K$). Both processes contribute with a median of ~5K (in 3 days) to the cross-isentropic flow. Additionally, the heated trajectories also experience cooling (probably evaporation of rain). By definition ($\Delta\theta > 2K$), the heating dominates. The convective contribution thus leads to an amplified ascent. We mention the contribution in the revised manuscript.

[Figure]

Figure **AR2**: Integrated temperature tendencies along heated trajectories ($\Delta\theta > 2K$) for the cloud microphysics (ttcloud) and convection (ttconv) scheme in the IFS: The box shows the median (orange), the 25–75% range and the bars the 5–95% percentiles.

11. l. 140 typo?
    **Reply** We corrected this, thank you.

12. l. 154–160 It is unclear why these specific cases were selected
    **Reply** See reply to general comment 2.

13. l. 165–167 For clarity, and because the nomenclature is used by Maddison et al.(2019),"Stalactite Cyclone" should be mentioned here
    **Reply** We corrected this, thank you.

14. l. 171–179 and Fig. 1 The paragraph needs improvement: (1) it is not "evident" to recognize the mentioned features, esp. at mesoscale (please zoom in and/or mark them);(2) a visual comparison between "upper-level" PV (defined as 500–150 hPa mean, please remind in the caption) and cloud top pressure (which is not directly "observed" by MSG) is not "quantitative"; (3) comparisons are for short lead times (36h and 42h,which should be indicated) thus do not support that the evolution is "well predicted" in the 10-day simulations and contradict the "large forecast uncertainty" mentioned above.
    **Reply** See reply to general comment 1. We changed the text and emphasized the qualitative nature of Figure 1. We marked features on Figure 1 and indicate the short forecast lead time to be more specific to that is shown in the figure.

**15.** l. 186 what is the APV "index" exactly?

    **Reply** The APV index is the PV anomaly-based blocking index by Schwierz et al., 2004. We improved the explanation in Section 2.3.1.

**16.** l. 190 "confirm": is it expected?

    **Reply** We added a reference to the climatological study by Steinfeld and Pfahl (2019), in which we analysed more than 4000 blocking events and showed that a block typically exhibits 2-3 bursts of LH, which are separated by periods of reduced LH contribution.

**17.** l. 191 why are quasi-adiabatic processes associated with cooling?

    **Reply** This is not explicitly shown; however, the quasi-adiabatic trajectories ($\Delta\theta < 2K$) are upper-level air masses that travel close to the tropopause and experience week cooling due to long-wave radiation along the flow. Figure AR3 shows the temporal evolution of pressure (a), $\theta$ (b) and the temperature tendency from radiation (c) along trajectories from all control simulations that have been separated into heated trajectories (yellow, $\Delta\theta > 2K$) and quasi-adiabatic trajectories (blue, $\Delta\theta < 2K$). The heated trajectories experience a median heating of ~10K and ascend by about 350hPa in 3 days, while the quasi-adiabatic trajectories are cooled by ~-3K in 3 days. This cooling can be explained by integrating the temperature tendency from radiation along the quasi-adiabatic trajectories, which results in ~-3K in 3 days.

[Figure]

Figure **AR3**: Temporal evolution of a) pressure, b) $\theta$ and c) temperature tendencies from radiation along the heated (yellow, $\Delta\theta > 2K$) and quasi-adiabatic (blue, $\Delta\theta < 2K$) backwards trajectories initialized in the blocking region (black dot at time 0). Lines show the median with 25-75% range (shaded) for trajectories from all control simulations (Thor onset, Thor maintenance, Canada, Cold spell and Russia).

**18.** l. 196–197 this is slightly below average compared to the climatology cited above

    **Reply** Yes, the median LH Contribution for 4270 blocking in the global ERA-I climatology is 45%. Thor has a slightly weaker LH contribution.

**19.** l. 207–208 is this shown somewhere? It is not obvious...

    **Reply** The mature phase (stable dipolar configuration) of block Thor lasts from 5 Oct (Figure 3c, Thor onset simulation) to 11 Oct (Figure 4a, Thor maintenance simulation). We explain this more clearly in the revised manuscript. However, we think that it is not necessary to show all time steps in between in the manuscript.

**20.** l. 210–219 The discussion is hard to follow, as the ingredients are not explicitly shown("jet splitting", "deformation region", "poleward transport", "ex-tropical cyclone", "migra-tory ridge"). Either detail and add information on Fig. 4, or streamline.

**Reply** Thanks, these comments are all valuable to improve the text and figures. See reply to general comment 3.

21. l. 226 how many is "many"?
**Reply** We replaced "many" with the fraction (~15%) of WCB trajectories in the manuscript.

22. l. 230 "Fig. 3 a, c, e, g": do you mean Figs. 3 a, c and 4 a, c? Only 3a and 4a are related to trajectories shown in Fig. 5a, c.
**Reply** Thanks for spotting this mistake, this is corrected in the revised manuscript. Note that also Fig. 3c and 4c show latent heating and divergent outflow aloft, which can be used as an indicator for WCB activity.

23. l. 241 see comment above
**Reply** Corrected.

24. l. 245, 247 "quasi-adiabatic": did you explicitly check the 2K heating criterion or do you refer to the stable pressure along trajectories?
**Reply** We explicitly calculate the changes in theta (heating and cooling) along the trajectories. Trajectories are characterised as quasi-adiabatic if their $\Delta\theta$ is < 2K. We make this clearer in the revised manuscript. See also the temporal evolution of pressure and $\theta$ in Figure AR3a in reply to comment 17.

25. l. 256 "mid-level": better lower-level in contrast with upper-level for 500–150 hPa?
**Reply** We replaced "mid-level" by "lower-level"

26. l. 258 "initial time steps" is confusing for day 2: better early evolution?
**Reply** We replaced "initial time steps" by "early evolution"

27. l. 270 "cold front" and l. 272, 285 "jet stream": are these features shown somewhere?
**Reply** See reply to general comment 3. Cold front (and lower-level temperature) is not shown in the manuscript. However, diabatic heating in Figure 3 indicates the position of the cold front.

28. l. 285 "as a consequence": I am not sure this is due to R2 only as the wave pattern is modified altogether (see l. 280)
**Reply** The analysis of the upper-level PV field with 3-hourly resolution allows for tracing the evolution of large-scale PV features (ridges) and the corresponding air masses (backward trajectories) and shows the contrasting evolution of R2 between CNTRL and NOLH simulation (see Figure AR4). In the control simulation (Fig. AR4a), R1 merges with the amplifying ridge R2, leading to a west- and equatorward extension of the blocking region. Without LH (NOLH simulation in Fig. AR4b), R2 does not amplify and merge with R1, but is advected eastward by the westerly winds. Instead, R1 is cut off from the tropospheric reservoir and surrounded by stratospheric air with high PV, leading to the formation of a cut-off anticyclone.
However, we think that it is not necessary to show all time steps in the manuscript, which would require a lot more panels.

[Figure]

a) CNTRL    b) NOLH

PV [PVU]
0.0 0.5 1.0 1.5 2.0 2.5 3.0 3.5 4.0 5.0 6.0 8.0

p [hPa]
200 300 400 500 600 700 800 900

Figure **AR4**: Upper-level PV (gray shaded, pvu), backward trajectories (colors in pressure, hPa) and blocking region (violet) for a) control and b) NOLH simulation at 00 UTC 6 Oct 2016. Black circles show the location of the backward trajectories 3 days prior to arrival in the blocking.

29. l. 291, 295 "deformation flow", "diffluent flow": not shown?
    **Reply** See reply to general comment 3. Because we focus on the large-scale dynamics, we use the z500 contours as a (geostrophic) approximation of the mid-tropospheric stream function/streamlines to describe the flow.

30. l. 296–297 see l. 285
    **Reply** See reply to l. 285

31. l. 298 is cooling explicitly computed along trajectories?
    **Reply** We indeed calculated cooling along the trajectories (this is mentioned in the revised manuscript). Moreover, we attached a Figure showing the statistical distributions of $\Delta\theta$ during the three-day backward trajectories for all five cases. The distributions reveal that the flow is actually never perfectly adiabatic. Following Steinfeld and Pfahl (2019), a threshold of $\Delta\theta = 2K$ is used to separate the blocking air masses between heated ($\Delta\theta > 2K$) and quasi-adiabatic ($\Delta\theta < 2K$) trajectories. Most quasi-adiabatic trajectories experience cooling of 3–4K in 3 days.
    Percentages of blocking air parcels in the heated flow regimes is given for control (gray) and NOLH (yellow) simulations.

[Figure]

Figure **AR5**: Probability density distribution of maximum potential temperature change along backward trajectories during three days before their arrival in the blocking region for control (gray line) and NOLH (yellow line) simulations. Percentages of blocking air parcels in the heated flow regimes defined by $\Delta\theta$ = 2K are given.

**32.** l. 301 Fig. 4 c, d
**Reply** Thanks for spotting the mistake.

**33.** l. 311–312 see l. 270, 291, 295,...
**Reply** See reply to general comment 3.

**34.** l. 322–323 again, what is the motivation for selecting these specific cases?
**Reply** See reply to general comment 2

**35.** l. 331–332 not really: (1) there is substantial case-to-case variability and (2) differences cannot be attributed to "the upstream cyclone" if it is neither showed nor mentioned for the additional cases
**Reply** The location of the upstream cyclones are now indicated by showing SLP in Figs. 3 and 4 and S1-S3. We also revised the discussion about the case-to-case variability (see reply to general comment 2).
An example: "Despite differences in the large-scale configuration, all cases show that the 2pvu contour is less amplified when LH is turned off. The biggest difference between CNTRL and NOLH occurs in all cases at the downstream side of an amplifying cyclone. The cases demonstrate that strong LH embedded in the upstream cyclone is crucial for the initial ridge amplification."

**36.** l. 339–342 is this all shown somewhere or suggested only?
**Reply** The divergent wind in the CNTRL simulations is not explicitly shown, but the difference between CNTRL – NOLH.

**37.** Fig. S4 does not include the additional cases.
**Reply** We show now PV advection by the divergent wind for all cases in Figure 9.

**38.** l. 350 where is the dipole pattern? Please indicate (a), (b), (c), etc.
**Reply** We now also show the centre of mass of the tracked anticyclones in Figure 7 and 8, which helps to better identify the features of interest and +ve and -ve area. We changed the wording "dipole pattern" to "positive and negative upper-level PV difference in the upstream and downstream troughs indicate a shift in location…" and reference to the Figure.

**39.** l. 355–361 this description (and the related panels) must be moved to the Thor Section above.
**Reply** We agree that Section 4 repeats some discussion about Thor before proceeding with a systematic analysis of all cases. We try to minimize repetition; however, we decided to keep this analysis here as it is related to Figure 7 and 8.

**40.** l. 369–376 That is certainly interesting but I do not know where to see all of it in Fig. 8.
**Reply** We now also show the centre of mass of the tracked anticyclones in Figure 7 and 8, which helps to better identify the discussed features (cut-off anticyclone versus dipole block)

**41.** l. 380–382 Does the box move with the strongest ascent/divergence/advection? Canyou show an example? On Fig. 8 in particular it is not obvious where it would beplaced.
**Reply** Yes, the box is placed at the western edge of the tracked blocking region and moves with the anticyclone. After consideration, we decided to replace Figure 9 with maps showing PV advection by the divergent wind for all cases (similar to S4) during onset / ridge amplification. Comparing the temporal evolution of ascent/divergence/advection, which are noisy fields, is challenging because the flow develops differently between CNTRL and NOLH (e.g. the box in CNTRL is not at the same geographical location as the box in NOLH).

**42.** l. 383–385 again, there is substantial variability both between cases and between lead times (weak signal beyond one week for instance)
**Reply** See reply to comment 41.

**43.** l. 385 "magenta": rather violet?
**Reply** We replaced " magenta" by " violet"

**44.** l. 386–387 is this shown somewhere?
**Reply** We improved the discussion about case "Russia" (the anticyclone propagates from Western Europe (day 2) to Russia (day 6).

**45.** l. 396 sorry to insist but there is again case-to-case variability: the Canada case does not show bursts
**Reply** We improved the discussion about case Canada and highlight that in contrast to the other cases, the Canada block is associated with only one (slowly moving) upstream cyclone with the largest LH contribution during onset that gradually declines to the lowest values when the block decays.

**46.** l. 396–409 the discussion is not supported by any material: cyclones are not shown in the figures but for the case of Thor
**Reply** See reply to general comment 3.

**47.** l. 401 which trajectories exactly? Cooling is not explicitly shown in Fig. 5.
**Reply** Cooling is not explicitly shown in Figure 5, but in Figure 10b. We improved the discussion about heating/cooling along trajectories. See also reply to comment 31.

**48.** Fig. 2 spatial "extent"; labels T1–T5 appear to refer to troughs rather than cyclones.
**Reply** Thanks for the suggestions; the figure was improved accordingly.

**49.** Figs. 7 why focus on day 3 here and not on day 2 as above?
**Reply** There is a temporal lag between the strongest difference in divergent outflow (day 2) and distinct differences in the upper-level PV (day 3). We show the same time step for all Figures in the revised manuscript. We also add a sentence stating the motivation for this selection.

**1 Response to Oscar Martinez-Alvarado**

**Referee comment**

Building upon recent findings related to the importance of tropospheric latent heating on the development of atmospheric blocking, this contribution investigates from a numerical modelling point of view, the extent to which latent heating influences the development of atmospheric blocking and the cause-and-effect relationship involved in this influence. Understanding these processes in the atmosphere is critical due to the important effects that blocking has at the surface and on human activities. Thus, these are without doubt relevant scientific questions within the scope of WCD. For this investigation the researchers performed sensitivity analysis by varying latent heating in ad-hoc regions in numerical simulation of five cases, using the state-of-the-art ECMWF IFS and an advanced methodology based on atmospheric blocking tracking and trajectory analysis. Through their investigation they demonstrate in a convincing manner that atmospheric blocking features such as intensity, spatial extent and lifetime depend strongly on latent heating. However, they also showed that there is a large case-to-case variability. The paper is very well structured and written, and, in my opinion, the description of the methodology is sufficiently complete to allow their reproduction by fellow scientists. Therefore I recommend the article for publication in Weather and Climate Dynamics. I include a list of minor comments that could be considered by the authors to hopefully enhance the paper.

**Specific comments**

**50.** L66: How smooth are the physical temperature tendencies in the native resolution? I fit is not a smooth field, is it properly represented after the interpolation to the 1-degree horizontal resolution?
**Reply** We mention now more carefully that the temporal and spatial resolution of the fields are an uncertainty in the trajectory analysis. We attached a Figure AR6, which shows temperature tendencies from cloud microphysics and convection schemes during the intensification phase of the Stalactite cyclone on 2 October 2019. Latent heating indicates the position of the cold front and the bent-back front in the vicinity of the cyclone's low centre. The panels show that there is some variability on small scales in the heating and cooling tendencies.

[Figure]

[Figure]

Figure **AR6**: Vertically averaged (900 – 500hPa) temperature tendencies from (a) cloud microphysics and (b) convection schemes (shading, K in 3h), upper-level 2 pvu contour (black, pvu) and SLP (gray, hPa) at 00 UTC 2 Oct 2016 for the control simulation of Thor onset.

51. L80-81: Please cite the previous studies that the methodology in this study is being contrasted against? In which way is the new methodology different to the one in previous studies? Did they dampened latent heating everywhere in their domain?
**Reply** We explain the novelty of our method more explicitly (modification in pre-defined box) and refer to previous studies using a similar methodology (but applied in the entire model domain) in the revised manuscript.

52. L105-106: How was the blocking event for which latent heating was reduced and in-creased chosen? Is the case representative in any way especially after considering that large case-to-case variability reported in this study?
**Reply** See reply to general comment 2 of referee 1. We tried to cover the typical range of different blocking flow configuration (omega versus dipole) with different LH contribution that represent the majority of observed blocking cases in the global ERA-I climatology between 1970 – 2016 (Steinfeld and Pfahl, 2019), where 50% of all blocking events had a LH contribution between 35 - 55%. However, we agree that there are also blocking cases in the climatological analysis that show no LH contribution (0%) or a contribution above 70%. It is just not possible to cover this entire range with a limited number of case studies.

53. L279-280: Is there any indication of the extent of the influence of initial conditions on the differences after 6 days? How would the differences found here compare to differences between members in an ensemble simulation? This is discussed to a certain extent in Maddison et al. (2020, doi:10.1002/qj.3739), which is in any case a relevant reference that you might want to cite.
**Reply** We include a reference to this important study in the revised manuscript. Evaluating the sensitivity to LH in ensemble simulations as performed by Maddison et al. (2019; 2020) would definitely be interesting, but is beyond the scope of the present study, which focuses on the causal effects of LH and not on predictability aspects. In our simulations, the initiation time was chosen such that the block is well captured in CNTRL, which is in contrast to Maddison et al. (2019; 2020). See also reply to general comment 1 of referee 1.
Since the removal of LH in our experiments are very pronounced changes to the simulations (in comparison also to differences in additional conditions and physical parameters between different members in an ensemble forecast), we assume that

also the differences between our experiments are larger than typical differences between ensemble members. Note that the differences presented in this study are shown for vertically averaged PV and that differences on a single level (for example PV@320K) are larger in magnitude.

54. L385-389: The Russia block is very interesting, and the discussion could be extended. If there is such a limited influence of latent heating in the evolution of the block, what is then the source of the big differences in the evolution of the blocks in the two simulations?
**Reply** We realized that we did not formulate this sentence carefully enough. The differences are due to changes in LH (since this is the only difference between CNTRL and NOLH). However, in both simulations the block propagates downstream, away from the heating source over the North Atlantic, into a continental region with weak LH contribution. Thus, the dynamics underlying the propagation of the Russia anticyclone after its onset is mostly due to "dry" dynamics and may therefore be understood with the help of the traditional concept of downstream development (Nakamura et al., 1997).
We extended the corresponding discussion, also about the other blocking cases (see reply to general comment 2 for referee 1).

55. L411: I've got a bit confused with this description, in which the authors talk about a median heating of 3 K (dashed curves in Fig. 10a,b). What I can see is cooling in those curves? I'm sure I'm missing something. Can you clarify?
**Reply** We are sorry for the confusion. The median heating of 3K is calculated for the entire blocking life cycle and only for those trajectories, which are classified as heating trajectories ($\Delta\theta > 2K$). The dashed curves in Fig. 10a,b show the temporal evolution of the median for all trajectories (quasi-adiabatic and heating trajectories). We improve the explanation of the changes in Theta along the trajectories in the revised manuscript (see also our reply to comment 31 of referee 1).

56. L435-436: Should the statement that the Thor onset and Cold spell block amplify without the contribution of LH be qualified? The LH was eliminated only between 900 - 500hPa, and as the authors acknowledge in Section 2 there are diabatic processes active above that layer.
**Reply** We agree that this statement is misleading. LH contribution is reduced, but it is not zero (see dashed curves in Fig 10a). We now discuss the limitation of the box and that diabatic processes still occur outside of the box in the NOLH simulations.

57. L464: Should the intensity of the upstream cyclone be included in the list of factors as is done in L432-433?
**Reply** Yes, you are right. Thanks for the comment.

**Technical corrections**

58. L58-59: Delete 'exemplarily' or change it for 'as an example' after 'introduces'
**Reply** Done

59. L65: In addition to the number of vertical levels, give details on the top of the atmosphere and the typical separation between levels.
**Reply** Done

**60.** L115: Change 'quasi-stationary' for 'quasi-stationarity'
**Reply** Thanks for spotting the mistake.

**61.** L123: Spatial extent is among the set of blocking characteristics calculated for each blocking event. Even though the method to identify blocking considers an atmosphere's layer rather than a single level, the extent referred to here is horizontal extent rather than a three-dimensional size. Is this so? It would be useful to add details on the layer considered for the blocking identification. Is it the 'upper-level' layer, i.e. 500 - 150hPa? Is it possible to compute details on the vertical extent of the blocking region?
**Reply** Yes, the blocking index uses vertically averaged (500 – 150hPa) PV as 2d field, which tracks negative PV anomalies that have a quasi-barotropic structure in the vertical. The spatial extent is then calculated as the horizontal extent of this 2d field.

**62.** L230 and 241: There are references to Fig. 3e,g, but I cannot see those panels.
**Reply** Thanks for spotting the mistake.

**63.** L220-221: Where are the trajectories emanating from in the vertical direction? Are they initially located between 500 hPa and 150 hPa? Or at a particular level?
**Reply** Yes, trajectories are started between 500 hPa and 150 hPa every 50 hPa, with the additional criterion that PV must be smaller than 1 pvu (to exclude points located in the stratosphere). We make this clearer in the revised manuscript.

**64.** L269: Are the divergent wind speeds quoted averages over a region? Please, specify.
**Reply** The quoted wind speeds are the average over 9 grid cells centred around the strongest divergent wind found at the western flank of the block. We make sure this is clear in the revised text. Note that we changed the discussion on divergent wind and PV advection by the divergent wind (see reply to comment 41).

**65.** L392: Add 'simulations' after '... NOLH (dashed lines)'.
**Reply** Done

[revised manuscript text omitted]

---

## Author Response (AR2)

**Author response to referee reports in manuscript wcd-2020-5:**

**"The sensitivity of atmospheric blocking to changes in upstream latent heating – numerical experiments"**
**by Daniel Steinfeld, Maxi Boettcher, Richard Forbes, and Stephan Pfahl**

We would like to thank both reviewers, Oscar Martinez-Alvarado and Florian Pantillon, for their second review and the detailed feedback to the manuscript. We have revised the manuscript accordingly (see below – our response in blue).

**1 Second review by Florian Pantillon**

**General comment**

I fully understand that the predictability of blocking situations is beyond the scope of the paper. The authors justify that "the reference simulations capture large-scale blocking conditions", which is true, but how close are these to the actual large-scale blocking conditions? If the model completely diverged from reality—which does not appear to be the case here—there would be no point at investigating real case studies. In their response figure AR1 the authors show that the control simulation is very close to the analysis after 3 days, which is good and strengthens the results, and deviates after 6 days, which is okay but deserves some discussion. This must be mentioned at least, and showed in any case, otherwise it may be interpreted as trying to hide some findings. As I suggested in my previous review, the analysis could be included by adding panels (or simply the 2-pvu contour) in Figs. 3–4 and S1–3 (or alternatively 7–8) and/or intensity and spatial extent in Figs. 2 and 10.

**Reply** We followed this recommendation and included a new Figure S4 with a brief discussion comparing the control simulations to ERA-Interim in the supplement. A reference to the supplement has been added in the manuscript: "Although a 10-day forecast simulation does not perfectly match observations/analysis (cf. supplemental Fig. S4), especially during blocking situations with increased forecast uncertainty (e.g., Tibaldi and Molteni, 1990; Pelly and Hoskins, 2003; Matsueda, 2009), such differences do not affect the conclusions obtained from the sensitivity experiments, since we compare simulations with LH (CNTRL) to simulations without LH (NOLH)."

Supplement 2: Synoptic comparison between CNTRL and ERA-Interim

The following Figure S4 compares upper-level PV between the control simulation (CNTRL) and ERA-Interim (ERA-I, Dee et al., 2011) for the cases Thor onset, Thor maintenance, Canada, Cold spell and Russia. After 3 days of model simulation (left panels in Fig. S4), the initial ridge amplification is very well represented in all cases with only minor differences in the upper-level PV compared to ERA-I. 6 days into the model simulation during the mature phase (middle panels in Fig. S4), the intensity and spatial extent of the mature block are generally well represented in the CNTRL simulations. However, positive and negative upper-level PV differences between CNTRL and ERA-I are found near the flanks of the blocking anticyclones, indicating a shift in location of the negative PV anomalies. 10 days into the model simulation (right panels in Fig. S4), both the CNTRL simulations and ERA-I show the decay phase of the blocks, but up- and downstream ridges and troughs are strongly displaced with marked differences in the upper-level PV pattern. Nevertheless, the forecast evolution of the blocks in the CNTRL simulations is similar enough to ERA-Interim over the time of interest (onset and mature phases) and captures an intense blocking anticyclone. It

thus allows studying the impact of LH on the flow amplification in the IFS sensitivity experiments.

[Figure]

**Figure S4**. Difference (CNTRL - ERA-I) in upper-level PV (shaded in pvu) and upper-level 2 pvu contour (solid for CNTRL, dashed for ERA-I) after (left panels) 3 days, (middle panels) 6 days and (right panels) 10 days of model simulation.

**Specific comments**

1. l. 34 "concepts": studies?
   **Reply** We replaced "concepts" with "studies".

2. The l. 83–99 specify the boxes are fixed during each run
   **Reply** We changed the sentence to "In order to isolate the effect of this LH, a 3-dimensional box is placed over the main heating region, which is kept fixed during each NOLH simulation, and LH is only modified in this box."

3. l. 105–107 this is not convincing…
   **Reply** See our reply to general comment.

4. l. 116–117 emphasize this terminology is used for all variable throughout the paper
   **Reply** Thank you. We added: "The term "upper-level" is used hereafter to describe the vertically averaged flow between 500 and 150 hPa."

5. l. 229–232 avoid referring to Fig. 1b as it does not show the features mentioned here)
   **Reply** We removed the reference to Fig. 1b.

6. l. l. 352 day 2 or 3?
   **Reply** Thanks for pointing out the mistake. Figure 6a shows 2 pvu contours on 3 October 2016 (day 3).

7. l. 356 "underestimated": weaker?
   Reply We replaced "underestimated" with "weaker".

8. l. 366–368 "demonstrates" is too strong, as uncertainty can arise from low-level moisture in the initial conditions, phasing between lower and upper-level flows, etc.
   **Reply** We replaced "demonstrates" with "suggests": "This strong sensitivity of block development to changes in upstream latent heating further suggests that forecast uncertainty during blocking can arise from diabatic heating from parametrized processes (e.g., Grams et al., 2018; Maddison et al., 2019)."

9. l. 438–547 (4.2.2) Fig. 9 is (1) very similar to Fig. 7, (2) only once referred to, and (3) implies returning to day 3 after discussing day 6 in Fig. 8. I thus recommend merging Fig. 9 with Fig. 7 (or alternatively with Figs. 3–4 and S1–S3) by simply adding a contour of strong PV advection by the divergent wind (see e.g. Fig. 7 in https://doi.org/10.1002/qj.2419) and merging contents of 4.2.2 with 4.2.1.
   **Reply** Thanks for your suggestion. We favor keeping 4.2.2 as a separate section. Adding PV advection by the divergent wind to Fig. 7, this and the corresponding discussion would become too complex and we think that the current version is easier to read.

10. l. 501–503 The definition of intensity and spatial extent indices is unclear: relative or normalized differences? Better express as percentage?
    **Reply** Thanks for spotting this mistake. Figure 11 shows the "normalized" differences:
    $$\frac{NOLH - CNTRL}{CNTRL}$$

11. Figure 1 "2 days lead time": rather 42h in (a) and 36h in (b)?
    **Reply** Thanks for the careful reading. We changed this.

**12.** Figure 11 use different symbols or annotate reduced and enhanced LH.
**Reply** Thanks for the suggestion. We changed the markers to better indicate the LH contribution for each case.

[Figure]

**Figure 11**. Normalized difference in peak spatial extent and peak intensity of the NOLH blocks compared to the CNTRL blocks. Values close to zero indicate weak sensitivity. Markers indicate the LH contribution in the CNTRL simulations (see Table 1). Red open circles for Thor onset simulations with reduced LH (α= 0.5) and enhanced LH (α= 1.5).

**2 Second review by Oscar Martinez-Alvarado**

**Specific comments**

**13.** Throughout the text (L2, L50, L77): This comment is more on semantics than atmospheric dynamics. While I agree with the objectives of the article to find a causal relationship between LH and blocking development, I would recommend changing the expression 'causal effect' for 'effect'. The modifier would only be required if there was an alternative type of effects, which I'm struggling to find.
**Reply** Thank you very much for the comment. We replaced 'causal effect' with 'effect'.

**14.** L80-81: L508-509: I'd rephrase the last part 'for which alpha=1.5 shows a stronger sensitivity…'. This gives the impression that sensitivity is a property of each LH setting. In my view, sensitivity is a measure of the changes in the target given changes in a parameter, LH in this case. My interpretation of these results would be that the sensitivity of blocking in the onset case is not linear with respect to changes in LH (as the curve joining the red dots must go through the origin). This is very interesting as it shows that changes in LH have an even greater effect on spatial extent rather than intensity as LH increases. It leaves the question of what happens in between the points shown open.
**Reply** That's right. The sentence was adjusted accordingly: "In addition, the Thor onset simulations with reduced LH (α = 0.5) and enhanced LH (α = 1.5) are shown as open red circles, highlighting that the sensitivity of blocking in the Thor onset case is not linear with respect to changes in LH. It shows that an increase in LH has an even

greater effect on spatial extent than on intensity, as blocking area increases by a value of 0.7 (by a factor 3) for α = 1.5."

**Minor changes and typos**

15. L90: Change 'diabatic PV modification at the tropopause' to 'the presence of diabatically modified PV at the tropopause'.
**Reply** We changed this.

16. L131: It is enough to say that the quantities are area-weighted. Therefore, you can delete 'with the cosine of latitude'. In fact the weighting function will depend on the data grid.
**Reply** We deleted this part.

17. L225: Change 'following Scherrer et al...' to 'the index described by Scherrer et al…'.
**Reply** We changed this.

18. L243: Change 'They…' to 'These trajectories…'
**Reply** We changed this.

19. L245-246: About the sentence starting with 'Both cloud microphysics…', as it is written I understand from that both schemes together contribute 5 K heating in total, whereas I think what you mean is that each scheme contributes about 5 K heating.
**Reply** Thanks for pointing this out. We adjusted this sentence: "Each scheme (cloud microphysics and convection) contributes about 5 K to the total diabatic heating along these ascending trajectories (not shown), …"

20. L253: I'd recommend changing 'underpins' for 'underlines' or 'highlights'. To me 'underpins' would imply that the importance of LH requires the trajectory behaviour described.
**Reply** We changed this.

21. L376: Perhaps 'While' would be more appropriate than 'Whereas', as there is no contrast between items here.
**Reply** We changed this.

22. L399: Add 'The' before 'largest differences'.
**Reply** Done

23. L441: I think it should read 'i.e.'.
**Reply** Thanks for spotting the mistake.

24. L510: Delete 'thus'.
**Reply** Done.

25. Figure 11: It would be good to add a quantitative indication of the LH contribution represented by the circles. Perhaps you could add a couple of reference circles?
**Reply** See reply 12 to comment for referee 1.